# Structure-forming CAG/CTG repeats interfere with gap repair to cause repeat expansions and chromosome breaks

Erica J. Polleys [1] ✉, Isabella Del Priore[1], James E. Haber [2] & Catherine H. Freudenreich [1] ✉

Expanded CAG/CTG repeats are sites of DNA damage, leading to repeat length changes. Homologous recombination (HR) is one cause of repeat instability and we hypothesized that gap filling was a driver of repeat instability during HR. To test this, we developed an assay such that resection and ssDNA gap fill-in would occur across a $(CAG)_{70}$ or $(CTG)_{70}$ repeat tract. When the ssDNA template was a CTG sequence, there were increased repeat contractions and a fragile site was created leading to large-scale deletions. When the CTG sequence was on the resected strand, resection was inhibited, resulting in repeat expansions. Increased nucleolytic processing by deletion of Rad9, the ortholog of 53BP1, rescued repeat instability and chromosome breakage. Loss of Rad51 increased contractions implicating a protective role for Rad51 on ssDNA. Together, our work implicates structure-forming repeats as an impediment to resection and gap-filling which can lead to mutations and large-scale deletions.

Alternative DNA structures formed by expanded CAG/CTG repeats can result in the formation of a barrier, making it difficult for both DNA replication and repair machineries to proceed smoothly. As such, these repetitive regions are hotspots for genomic change. When a breakage event occurs within a CAG/CTG repeat tract, it can be repaired by homologous recombination (HR), using a region of homology on a sister chromatid or homologous chromosome as a template. Though HR is generally considered to be an error-free mechanism of repair, the fidelity of repair through a CAG/CTG repeat may be compromised, as HR has been shown to be a mechanism for repeat expansions[1]. Additionally, in yeast strains containing a long CAG repeat tract on a yeast artificial chromosome (YAC), recombination-dependent expansions and contractions were observed in strains carrying mutations in proteins important in DNA repair or replication[2].

The mechanisms that drive repeat instability during HR are not fully understood. Key steps in HR involve the 5′ to 3′ resection of DSB ends and the subsequent filling in of ssDNA regions. Filling in ssDNA gaps is one point where CAG/CTG repeats could expand, as polymerases may be prone to slippage across the CAG/CTG repeat. Repair

DNA synthesis is 1000-fold more mutagenic than replication of the same sequences[3,4]. Polymerase slippage while filling in ssDNA gaps that arise during mismatch repair (MMR) was proposed to be responsible for germline repeat expansions in a Huntington's disease mouse model[5], but how gap filling proceeds in the context of a CAG/CTG repeat has not been directly determined. The processivity of the polymerase, size of the gap and stability of the DNA secondary structure could all contribute to trinucleotide repeat (TNR) instability during gap repair[6].

Resection is a highly conserved process that is considered one of the key steps that drives repair away from end-joining and toward HR. Resection is restrained by 53BP1[7–9] and it has been proposed that the 53BP1 ortholog in budding yeast, Rad9, forms a dynamic barrier at the ssDNA/dsDNA junction through interaction with Dpb11 and the phosphorylated form of histone H2A, γH2AX[10]. Outside of its role in resection, Rad9 has a well-characterized role in activation of the DNA damage checkpoint[11]. Previous work has shown that loss of Rad9 resulted in a significant increase in CAG repeat fragility and instability due to its role in checkpoint activation[12]. It remained to be determined

[1]Department of Biology, Tufts University, Medford, MA 02155, USA. [2]Department of Biology and Rosenstiel Basic Medical Sciences Research Center, Brandeis University, Waltham, MA 02454, USA. ✉e-mail: Erica.Polleys@tufts.edu; Catherine.Freudenreich@tufts.edu

whether repeat stability would be impacted by resection kinetics, and if so, whether Rad9 played a role.

In this work, we use an assay system that repairs an induced DSB via single-strand annealing (SSA)[13] to determine whether ssDNA gap repair results in CAG/CTG repeat instability. We find that the template of the repeat dictates the efficiency of repair kinetics as well as the type and magnitude of repeat instability. When a $(CAG)_{70}$ repeat tract is the template for gap filling we note no loss in expected repair, though resection is impaired and small-scale repeat expansions are observed. Conversely, when a $(CTG)_{70}$ repeat tract is the ssDNA template for filling in, there is a significant decrease in expected repair caused by breakage at the repeat tract and the recovered repair products contain large-scale deletions and repeat contractions. Increasing the rate of resection and thus the kinetics of RPA and Rad51 loading by deleting Rad9 reduces repeat contractions and increases viability. In contrast, deleting Rad51 results in increased repeat contractions, suggesting that Rad51 can function like RPA in preventing DNA secondary structure formation at expanded repeat tracts. Our data show that resection and gap filling through a repeat tract are key steps required to prevent repeat instability and protect genome integrity. This work illustrates that large ssDNA gaps create an ideal environment for DNA secondary structure formation, which can act as a fragile site to cause large-scale deletions. Therefore, the repeat content and structure-forming potential of the region surrounding a DSB can determine repair pathway choice and fidelity.

## Results

### Identity of the repeat on the template strand determines survival during gap fill-in

To determine whether fill-in synthesis resulted in CAG/CTG repeat instability, we integrated a $(CAG/CTG)_{70}$ repeat tract into a strain that repairs an induced DSB via SSA[13]. A DSB induced by the HO endonuclease within in the *LEU2* gene initiates resection on both sides of the DSB. Once 25 kb of resection occurs, a homologous region within the LEU2 gene (U2) is exposed, allowing for the U2 homologies to anneal. Finally, regions that were rendered single-stranded during resection are filled in by DNA polymerases (Fig. 1a). Successful repair is measured as percent viability. To study instability of the CAG/CTG repeat during fill-in synthesis, TNR tracts were integrated into the *ILV6* locus, a non-essential gene ~13 kb away from the DSB on the centromere-proximal side. Because resection occurs equally on both sides of the break[14], this distance ensured that the repeat tract would be fully single-stranded by the time the U2 region on the other side of the HO cut site was single stranded. This distance facilitated kinetic analysis as it provided time to follow both the resection and repair steps. We inserted repeat tracts such that either 70 CAG or CTG repeats were on the strand that serves as the template for DNA fill-in synthesis (Fig. 1b).

To be able to attribute any altered outcomes to the presence of the repeat tract, we created a scrambled control containing a non-structure-forming sequence with equal amounts of C, T, and G on the template (scrm$(CTG)_{70}$) (Fig. 1b). The scrm$(CTG)_{70}$ control has similar viability compared to the original assay strain (no repeat) (Fig. 1c)[13]. Monitoring of HO cleavage, 5′ to 3′ resection and U2 repair product formation, as well as the activation and extinguishment of the DNA damage checkpoint (i.e. phosphorylation of Rad53) showed no differences between the no repeat and scrm$(CTG)_{70}$ strains (Supplementary Fig. 1a–d); thus the addition of a non-repetitive sequence does not alter repair and is a satisfactory control for our repeat-containing strains. Interestingly, strains that had an inserted $(CTG)_{70}$ repeat on the fill-in template showed a significant four-fold decrease in viability compared to the scrambled control strain (Fig. 1c). Previous work has shown that TNR repeat fragility is length dependent, with longer repeats having higher rates of breakage[15]. To confirm that the loss in viability is due to repeat length, we constructed a $(CTG)_{30}$ assay strain and found no significant defect in viability compared to the scrambled

control strain (Fig. 1c). In addition, strains that had a $(CAG)_{70}$ fill-in template had no viability defect (Fig. 1c). The differences in viability between the $(CTG)_{70}$ and $(CAG)_{70}$ strains suggested that the template of an expanded repeat tract could influence repair outcome, and thus cell survival.

### Identity of the sequence on the template strand determines resection kinetics and repair efficiency of gap filling

We next explored whether the addition of an expanded repeat tract altered the kinetics or efficiency of resection and repair, and if this could explain the loss in viability seen in the $(CTG)_{70}$ template strain. Tracking DSB induction and repair product formation via Southern blotting[13] revealed that the expected U2 repair product still forms in the $(CTG)_{70}$ template strain with the same timing, though at a significantly reduced level, compared to the scrambled control or the $(CAG)_{70}$ template strain (Fig. 2a, b). One possible reason for the decreased viability could be due to persistent activation of the DNA damage checkpoint[13]; therefore, we assessed the state of Rad53 phosphorylation by Western blot after DSB induction. None of the strains showed persistent hyperactivation of Rad53, ruling out that the decreased viability in the $(CTG)_{70}$ template strain was due to a defect in recovery from checkpoint activation (Supplementary Fig. 1d).

As this assay relies on extensive resection to expose the homologous repair sequence (U2, Fig. 1a), it is an ideal system to monitor resection and subsequent gap filling. Using a qPCR-based assay to quantify levels of ssDNA[16,17], we monitored resection and fill-in kinetics 600 bp after the repeat tract (Fig. 2c). In the scrambled control strain, we saw increasing amounts of ssDNA appear between hours 2–6 and a subsequent decrease in ssDNA signal from hours 8–24 (Fig. 2b, black line). Though resection across this region is maximal between 2–6 h, double-stranded repair product formation can already be visualized 4–6 h post-DSB induction (Fig. 2a). Notably, there is an increase in the amount of ssDNA in the $(CTG)_{70}$ template strain compared to the scrambled control 6 h post-DSB induction; this increased ssDNA is significantly elevated at all remaining time points (Fig. 2b).

During resection, RPA is recruited to ssDNA and helps prevent DNA secondary structure formation[18]. In addition, Rad51 is recruited to ssDNA to initiate the formation of the nucleoprotein filament necessary for the homology search[19]. Though the resected break in this assay system predominantly repairs via SSA it can also repair via BIR which is Rad51-dependent[19]. With the expectation that RPA and Rad51 enrichment would increase during resection and then decrease during gap filling, we monitored enrichment of RPA and Rad51 in both the $(CTG)_{70}$ and scrm$(CTG)_{70}$ strains. Monitoring enrichment of RPA and Rad51 either 60 bp or 600 bp after the repeat tract showed increasing enrichment of both proteins up until 6 h post-DSB induction and then a subsequent decrease in enrichment, mirroring the timing profile of resection and fill-in. Comparing the $(CTG)_{70}$ tract to the scrm$(CTG)_{70}$ control showed no significant differences in the level of RPA and Rad51 enrichment (Supplementary Fig. 2a–d). Taken together, these data suggest that there is a defect in completing gap filling and repair when $(CTG)_{70}$ is the fill-in template that is not due to impaired recruitment of RPA or Rad51.

Even though there is no repair defect in the $(CAG)_{70}$ template strain (Fig. 2a, b), a decrease in ssDNA accumulation post-DSB induction was observed in this strain compared to the scrambled control, indicating that less ssDNA accumulates beyond the repeat tract when $(CTG)_{70}$ is on the resected strand (Fig. 2c). When we examined resection and filling in at a location before the repeat tract, we observed significantly less ssDNA in the $(CAG)_{70}$ template strain compared to scrm$(CTG)_{70}$ or $(CTG)_{70}$ strains at later time points, consistent with there being a smaller gap that is filled in more quickly (Supplementary Fig. 2e). In summary, whereas a CTG hairpin on the template strand leads to increased resection and delayed gap filling, a CTG hairpin on

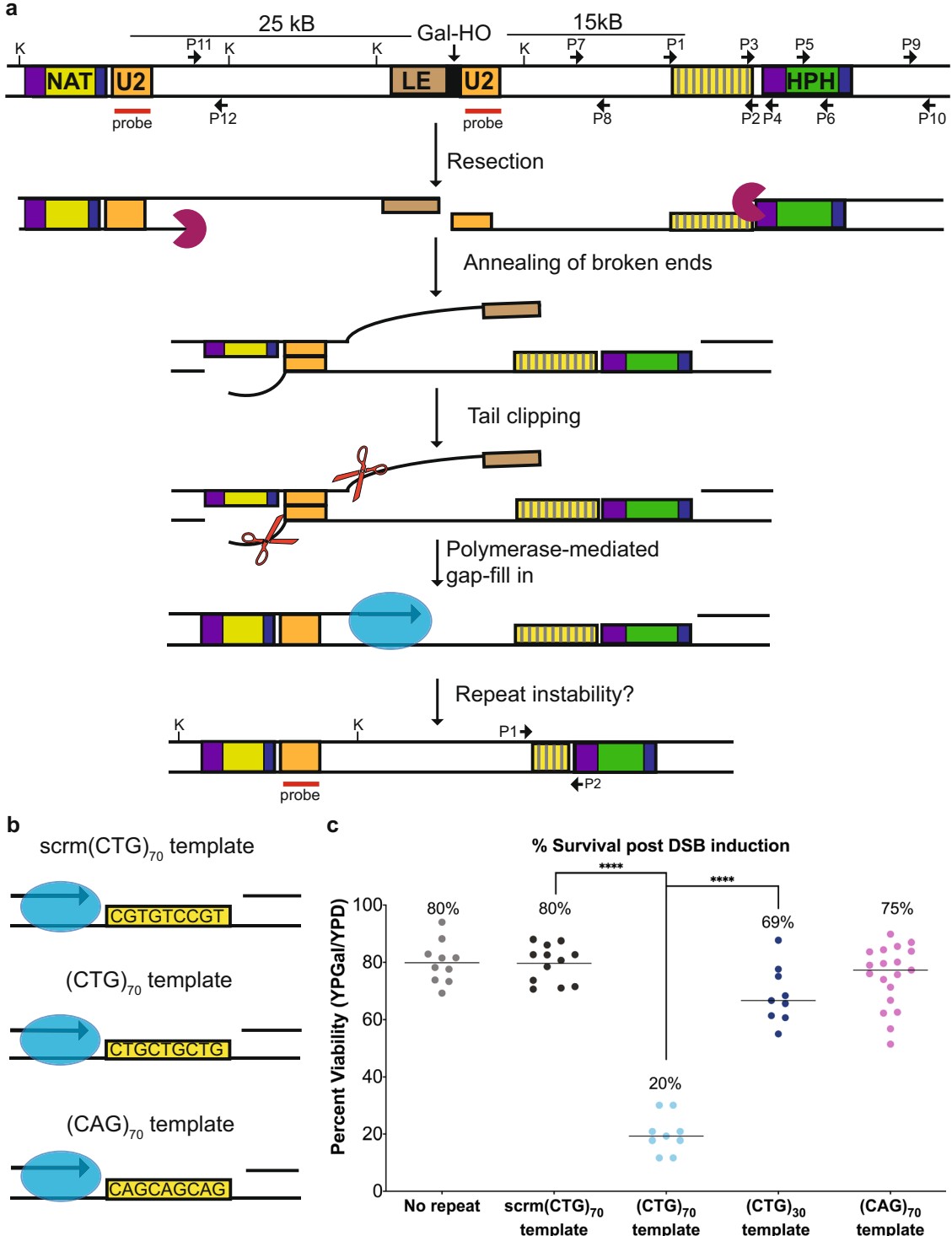

**Fig. 1 | Identity of the repeat tract on the template strand determines survival post-DSB induction. a** Assay system to study gap repair mediated CAG/CTG repeat instability. Modifying strain YMV80[13], we inserted a CAG/CTG repeat tract -13 kb centromere-proximal from an HO endonuclease cut site within *LEU2*. Following galactose-induction of the HO endonuclease, repair occurs by annealing of the two 1-kb U2 homologies. Fill-in synthesis occurs through the inserted repeat tract after U2 annealing. Relevant *Kpn*I sites are marked. Probe used for analysis of U2 repair is marked. P1-P12 primer locations shown; primer sequences are listed in Supplementary Table 1. **b** In this assay system, the template of the CAG/CTG repeat is defined as the sequence that remains after 5' to 3' resection of the DNA creates a single-stranded template. The fill-in template of the inserted repeat tract is either a scrambled control (scrm(CTG)$_{70}$), (CTG)$_{70}$ or (CAG)$_{70}$. **c** Viability (%) of the no repeat control (*n* = 10), scrambled (*n* = 12), (CTG)$_{70}$ (*n* = 9), (CTG)$_{30}$ (*n* = 9) and (CAG)$_{70}$ (*n* = 19) template strains where *n* represents assays from biologically independent experiments. For statistical comparisons, (CTG)$_{70}$ is compared to the scrambled control (*p* > 0.0001) and (CTG)$_{30}$ (*p* > 0.0001) using an unpaired, two-tailed Student's *t* test. Source data are provided as a Source Data file.

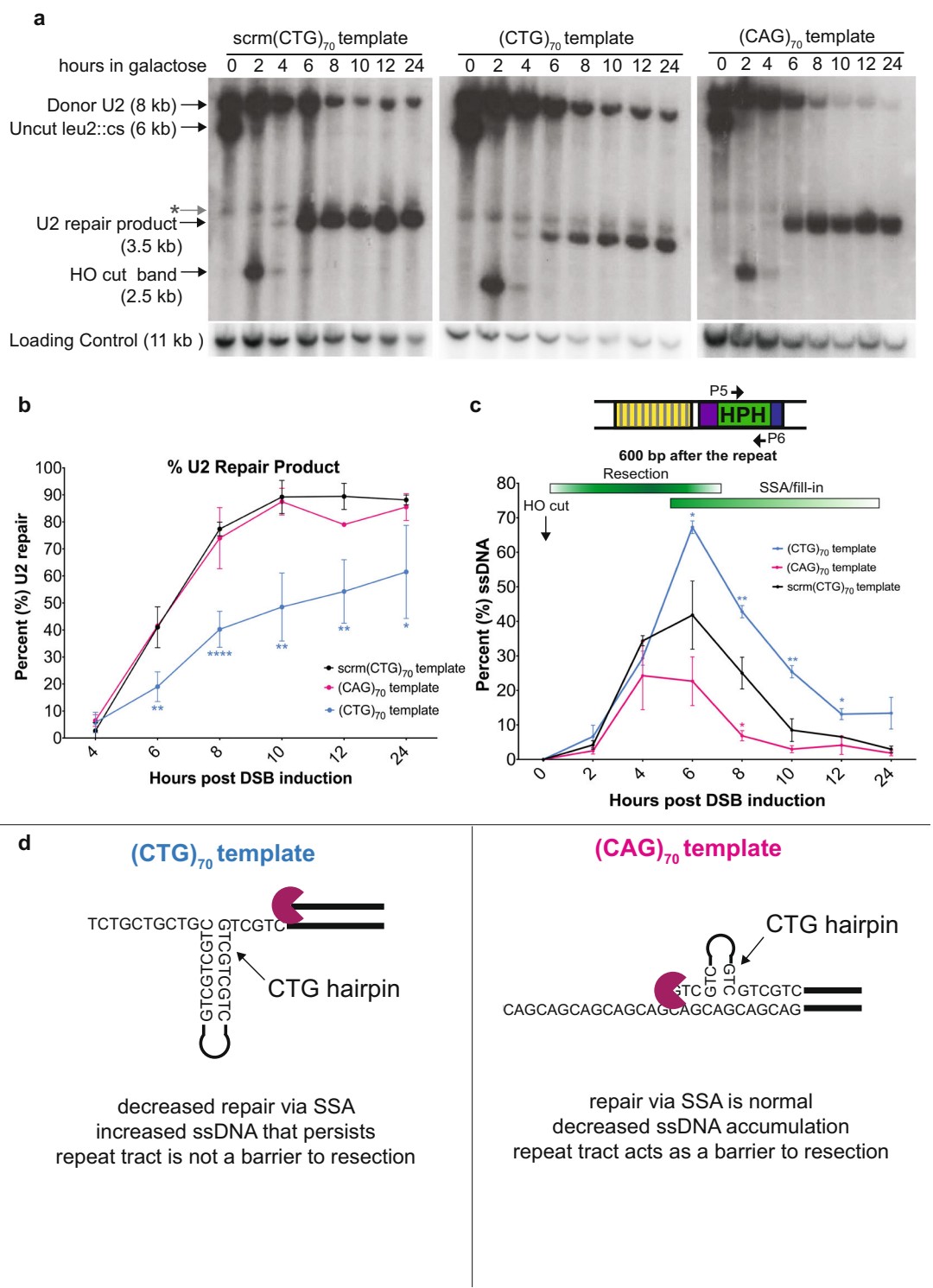

**Fig. 2 | Identity of the repeat on the template strand results in alterations in repair kinetics during gap filling. a** Southern blot analysis after addition of 2% galactose to induce a DSB within *LEU2*. Representative Southern blot is shown; number of replicates: scrm(CTG)$_{70}$ $n$ = 3, (CTG)$_{70}$ $n$ = 4, (CAG)$_{70}$ $n$ = 3. U2 probe location is marked in Fig. 1a; loading control probe is to a portion of the *TRA1* gene. Non-specific hybridization of the probe is marked with a gray star. **b** U2 repair measurement (%) on Southern blots after DSB induction. Number of replicates measured: scrm(CTG)$_{70}$ $n$ = 3, (CTG)$_{70}$ $n$ = 4, (CAG)$_{70}$ $n$ = 2 where n represents biologically independent time courses. Graph shows mean ± SD. Statistical significance determined by an unpaired Student's *t* test using a two-stage step-up with

a false-discovery rate of 1% (Benjamini, Krieger, and Yekutieli). Source data are provided as a Source Data file. **c** Formation and disappearance of ssDNA 600 bp after the repeat tract using primers P5 & P6. Number of replicates: scrm(CTG)$_{70}$ $n$ = 3, (CTG)$_{70}$ $n$ = 4 and (CAG)$_{70}$ $n$ = 3 where $n$ represents biologically independent time courses. Graph shows mean ± SD. Statistical significance determined using an unpaired Student's *t* test using a two-stage step-up with a false-discovery rate of 1% (Benjamini, Krieger, and Yekutieli). Source data are provided as a Source Data file. **d** Summary of results and model for resection and secondary structure formation in each template.

the resected strand leads to decreased resection and faster gap filling (Fig. 2d).

Monitoring ssDNA levels and repair kinetics suggested a model for how resection through the repeat tract influences repair and thus survival. In strains that have a $(CTG)_{70}$ fill-in template, there is increased, persistent ssDNA at the repeat tract (Fig. 2b) which coincides with a decrease in the expected U2 repair product and survival (Figs. 1c, 2a). These results suggest that CTG hairpin formation on the ssDNA template strand is impeding expected repair (Fig. 2d, left panel). In strains that have a $(CAG)_{70}$ fill-in template, we observed reduced ssDNA after the repeat tract (Fig. 2c) and faster gap filling whether measured before (Supplementary Fig. 2a) or after (Fig. 2c) the repeat. We propose that in this case, the CTG hairpin on the 5′ recessed strand acts as a barrier to resection, which decreases the size of the ssDNA gap due to the hairpin-impaired resection (Fig. 2d, right panel).

## CAG/CTG expansions and contractions occur during gap filling

Our assay system tracks changes in repeat length both in conditions where there is no induced HO break (in glucose) or post-induction of the DSB at *LEU2* (in galactose) as distinct populations. The instability of expanded repeat tracts can occur during normal DNA transactions such as DNA replication and repair. The direction of replication influences the basal stability of structure-forming repeats: CTG on the lagging strand template favors contractions, and CTG on the nascent lagging strand favors expansions[20–22] because CTG repeats can form a more thermodynamically stable hairpin compared to CAG repeats[23]. In addition, expanded repeats are fragile and prone to DSBs within the repeat which can result in out of register annealing events that lead to expansions or contractions[2]. Indeed, DSB induction using Cas9 to target an expanded repeat tract increases expansion and contraction frequencies[24,25]. The observed instability in the no-break condition could be due to either or both events. In contrast, the induced HO break condition is testing instability due to DNA synthesis during gap filling which occurs independently of instability due to replication or naturally occurring breaks within the double-stranded repeat.

To determine whether gap filling resulted in changes in TNR repeat stability, we used PCR to compare tract length changes in colonies from the no-break and HO-induced break (DSB) conditions using primers that span the repeat tract (Fig. 1a, Primers P1 & P2). Products were separated using capillary gel electrophoresis (Supplementary Fig. 3a, b) and the frequency of PCR product sizes of independent colonies tested was plotted (Fig. 3a, b). Expansions and contractions were defined as ≥3 bp above (E, expansion) or ≥3 bp below (C, contraction) the median determined in the no-break condition; regions between the lines are defined as unchanged (U) (Fig. 3a, b). In the assay system used here, the repeat tract was inserted 4 kb away from the ARS307 origin of replication which is to the right of the *HPH* gene (Fig. 1a). Thus, in the $(CTG)_{70}$ template strain, the CTG sequence is on the lagging template strand during replication. Consistent with previous work[20–22], this results in a high basal contraction frequency of 48% in the no-break condition. Interestingly, contraction frequencies significantly increase in the DSB condition to 78% (Fig. 3c). Separation on the fragment analyzer allowed for precise determination of how many repeat units were either gained or lost. In the $(CTG)_{70}$ no-break condition, the size of the contractions was between 1 and 64 repeats (3 to 193 bp), with a median loss of 41 repeats (123 bp). This contraction size range is somewhat larger in the break condition, where colonies lost between 1 and 68 repeats (3 to 205 bp) and had a median loss of 48 repeats (144 bp). There is no significant break-dependent change in repeat expansions in the $(CTG)_{70}$ template strain (no-break: 2.5%, break condition: 1.7%) (Fig. 3d). Together, these data show that gap filling increases contraction frequency and large-scale contractions are favored when CTG is on the ssDNA template strand.

In contrast, instability in the $(CAG)_{70}$ template strains shifted to larger (expanded) sizes after DSB repair compared to the no-break condition (Fig. 3b). Interestingly, most of these expansion events are small and only add 1 repeat (3 bp) to the repeat tract; 88% of expansions are within 1 repeat of the expansion cutoff (Fig. 3b). In the no-break condition, there is a 3.4% expansion frequency, whereas in the break condition, the expansion frequency significantly increases to 18.7%, a 5.5-fold increase (Fig. 3d) showing that gap filling is driving small repeat expansions. The shift towards expansions also resulted in a shift away from contractions when $(CAG)_{70}$ is the fill-in template (Fig. 3c). The contraction frequency is 19.2% in the no-break condition and significantly decreases to 5.5% in the break condition, a 3.5-fold decrease. The reduction in contractions supports that the no-break and break conditions are measuring instability due to different biological events, e.g. replication or repair of naturally occurring breaks within the repeat versus gap filling over a single-stranded repeat. To confirm that gap fill-in mediated instability was a function of the structure-forming ability of the repeat tract, we determined the instability of both the $(CTG)_{30}$ and the scrambled control strains (Supplementary Fig. 3c–f). Though some instability exists for $(CTG)_{30}$ and the scrambled control, expansion and contraction frequencies were not significantly different between the no-break and break conditions: $(CTG)_{30}$ (Supplementary Fig. 3d, f). Taken together, this suggests repeat length and ability to form DNA hairpins are prime factors that drive repeat instability during gap-filling.

The differences in expansion and contraction frequency between the $(CAG)_{70}$ and $(CTG)_{70}$ template conditions suggest the following model for the causes of CAG/CTG repeat instability during gap filling. In the $(CTG)_{70}$ template strains, there is impaired repair and increased contractions, but resection occurs unimpeded. The lack of resection defect could be explained by the fact that the CAG tract on the 5′ recessed end forms a less stable hairpin and is more easily unwound. On the other hand, the ssDNA exposed on the template strand by resection can form stable CTG DNA hairpins. Bypass of these hairpins during polymerase mediated fill-in could result in the frequent and large-scale contractions observed (Fig. 3e). Alternatively, a break that occurs within the single-stranded CTG template after annealing of the U2 homologies (so that gap filling provides a CAG top strand) could be repaired such that out of register alignment results in a repeat contraction (Supplementary Fig. 3g). In the $(CAG)_{70}$ template strains, our data indicate a difficulty resecting through the repeat tract, suggesting a barrier such as a CTG hairpin on the 5′ end of the resected strand. This hairpin could be resistant to processing by the canonical long-range DSB resection machinery (consisting of ExoI and Dna2/Sgs1 in yeast[14,26,27]) and/or other endo- or exonucleases. To explain the increase in gap fill-in-dependent expansions two models can be envisioned. One possibility is that the hairpin on the resected strand remains unresolved and is incorporated by ligation after polymerase gap filling (Fig. 3f). Another possibility is that expansions occur by polymerase slippage during gap filling (Fig. 3f). These possibilities are not mutually exclusive: slippage may be further promoted by the presence of a hairpin on the 5′ flap which could impair polymerase release and ligation. We cannot exclude that other models for repeat length changes could exist. Regardless, these data show that the template of a repeat tract with respect to repair synthesis determines its stability in the genome and whether it will be more prone to expansions or contractions.

## Breaks at the single-stranded CTG tract explain the decreased viability in the CTG template strain

There is a dramatic decrease in viability in the $(CTG)_{70}$ template strain (Fig. 1c). We hypothesized that this may be due to fragility at the CTG repeat tract post-DSB induction at the HO site, resulting in two breaks. The assay system we constructed contains a second duplicated sequence, as the MX cassettes share homologies on either side of the *HPH* and *NAT* drug-resistance markers (Fig. 4a, purple boxes). If a break at the single-stranded CTG repeat tract occurred, the MX homologies

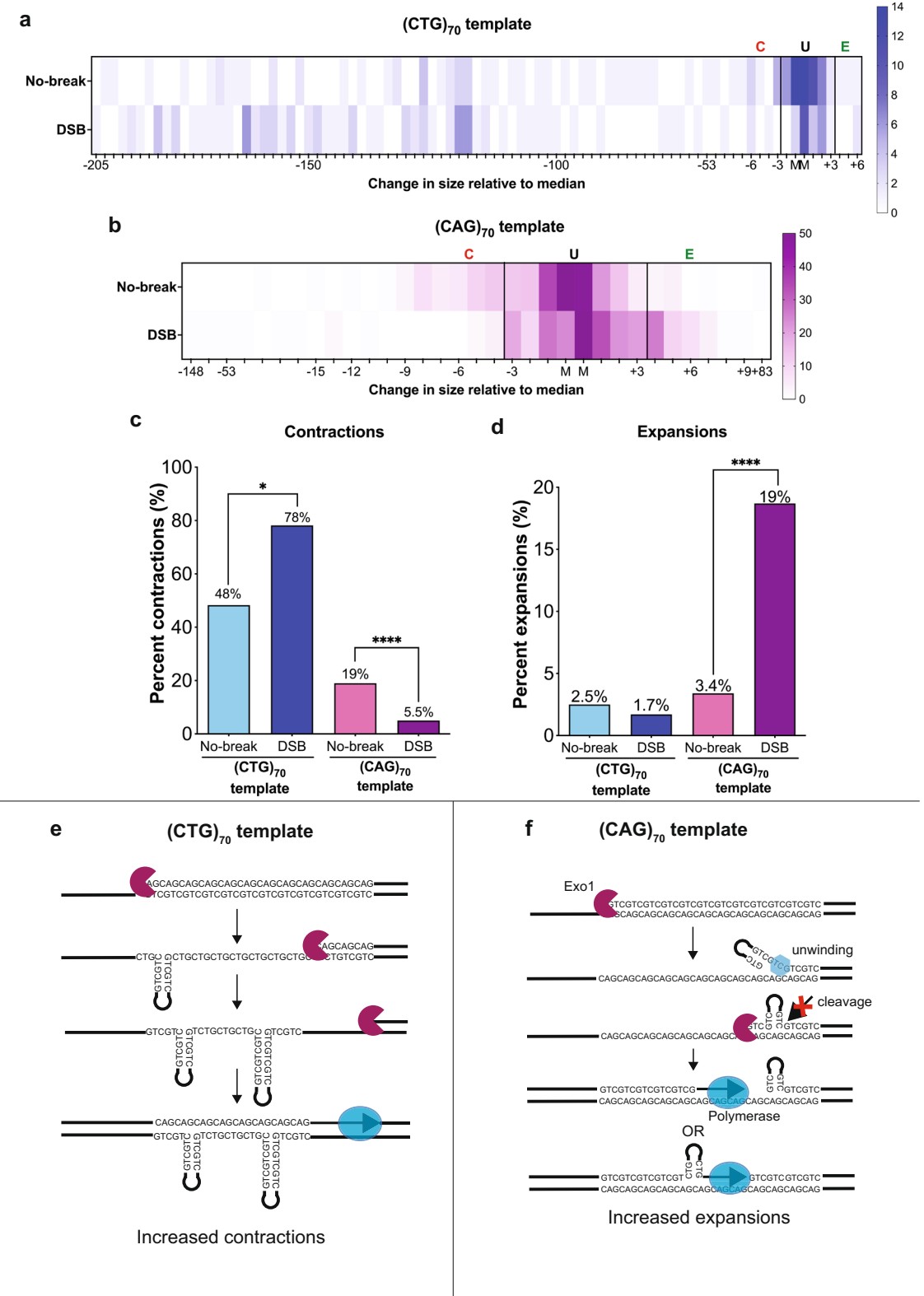

**c** Contractions

**d** Expansions

**e** (CTG)$_{70}$ template — Increased contractions

**f** (CAG)$_{70}$ template — Increased expansions

could be a site of recombinational repair; however, this larger deletion would also delete the essential *NFS1* gene and be inviable. To determine if this event was occurring, we tested whether supplying the cell with a copy of *NFS1* would be sufficient to rescue the viability defect. Using genetic complementation of *NFS1* on a single copy plasmid, we found that complementing *NFS1* resulted in a significant increase in viability of the (CTG)$_{70}$ template strain compared to the vector only control (Fig. 4b). In the (CTG)$_{70}$ + *NFS1* complementation experiments

there were two populations of colonies. The first grew as expected and had an amplifiable CTG tract (large colonies; Supplementary Fig. 4a). The second, which only appeared after *NFS1* complementation, grew poorly, and did not have an amplifiable repeat tract, suggesting these were the colonies in which the larger deletion had occurred (small colonies; Supplementary Fig. 4a). As the small colonies were difficult to propagate, it is likely that while complementation with *NFS1* is sufficient for rescuing viability, loss of additional genes on the region

**Fig. 3 | Gap filling during repair results in repeat instability. a** Heat map showing repeat length changes measured in the $(CTG)_{70}$ template strain. M denotes median tract length of no-break condition. Selected region between black lines is unchanged (U). Expansions designated as (E), contractions designated as (C). Tick marks denote size change (in bp) relative to the median. Number of PCR reactions represented: no-break n = 120, break n = 119. **b** Heat map showing repeat length changes measured in the $(CAG)_{70}$ template strain. Abbreviations are as in (**a**). Number of PCR reactions represented: no-break $n = 261$, break $n = 273$. **c** Contraction frequency was determined by counting lengths below the unchanged threshold determined in (**a**) and (**b**). Left: for the $(CTG)_{70}$ template, contractions significantly increased in the DSB condition ($n = 119$) compared to the no-break condition ($n = 120$), $p = 0.02$. Right: for the $(CAG)_{70}$ template, contractions significantly decreased in the DSB condition ($n = 273$) compared to the no-break condition ($n = 261$), $p = 0.0001$. **d** Expansion frequency was determined by counting lengths above the unchanged threshold in (**a**) and (**b**). Left: for the $(CTG)_{70}$

template, expansion frequencies are not significantly different between the no-break ($n = 120$) and DSB conditions ($n = 119$). Right: for the $(CAG)_{70}$ template, expansions significantly increased with DSB induction ($n = 273$) compared to the no-break condition ($n = 261$), $p = 0.0001$. For (**c**) and (**d**) each $n$ value represents PCR of an independent colony, statistical analysis by Fisher's exact test. **e** Model for $(CTG)_{70}$ template repeat contractions during gap filling. Resection over the $(CTG)_{70}$ template occurs unimpeded. The resulting ssDNA is left unprotected such that DNA hairpins can form. Polymerase fill-in bypasses hairpins resulting in repeat contraction. **f** Models for $(CAG)_{70}$ template repeat expansions during gap-filling. During resection, helicase unwinding or strand displacement results in a ssDNA flap on the strand being resected that forms a small CTG hairpin which is resistant to endonuclease cleavage. Failure to resolve the hairpin on the resected strand results in incorporation of that sequence during the restoration of the dsDNA molecule. Alternatively, polymerase slippage through the repeat tract results in the addition of bases. For figure (**a**–**d**) Source data are provided as a Source Data file.

between the HO site and the repeat tract predicted, such as the RFC protein Dcc1, also impairs cellular fitness.

This assay system can utilize both SSA and BIR repair pathways to complete repair[19]. If recombinational repair mechanisms like BIR were occurring between the MX cassettes, there are two possible homologies that are present in the *NAT* and *HPH* marker genes that could be used: the TEF promoter (344 bp) or the TEF terminator (198 bp). Recombination between the TEF promoters would result in the loss of the *NAT* marker gene, while recombination between the TEF terminators would result in the loss of the *HPH* marker gene. To test whether alternative recombination between markers could explain the loss in viability, we replica-plated all colonies on media containing either nourseothricin or hygromycin. As expected, 100% of large colonies contained both the *NAT* and *HPH* markers (Fig. 4c) indicative of repair using the expected U2 homology (Fig. 4a, left pathway). However, small colonies only contained both markers 8.7% of the time (Fig. 4c), consistent with alternative repair occurring most of the time (Fig. 4a, right pathway). Small colonies retained the *HPH* marker 64% of the time and the *NAT* marker 22% of the time, consistent with more recombination occurring at the TEF promoters that share more homology.

If the $(CTG)_{70}$ ssDNA forms stable hairpins, they could be a substrate for nucleolytic cleavage resulting in a second break[28]. To see if a second break could be detected, the Southern blots monitoring U2 repair from Fig. 2a were re-hybridized with a probe to the *HPH* gene 600 bp downstream from the repeat. We observed a unique band in the $(CTG)_{70}$ template strain that corresponds to the size expected if the break occurred within the $(CTG)_{70}$ repeat tract (Fig. 4d). Measurement of the signal of the break at the $(CTG)_{70}$ repeat showed an increase compared to the scrm$(CTG)_{70}$ at hour 4 and 6 post-DSB induction (Fig. 4d). This break was not observed for the CAG template or no repeat strains (Supplementary Fig. 4b). In addition, a band corresponding to the size expected for recombination between the TEF promoters was observed in the CTG repeat strain (Fig. 4d). Recombination between TEF promoters is also observed in the Scrm$(CTG)70$, $(CAG)_{70}$ and no repeat strains (Fig. 4d; Supplementary Fig. 4b), however promoter recombination was increased in the $(CTG)_{70}$ template strains compared to the scrm$(CTG)_{70}$ control (Fig. 4d, Supplementary Fig. 4c). The break at the $(CTG)_{70}$ template strain appears at hour 4, suggesting the repeat tract breaks when the DNA in this region has become single-stranded (Fig. 4d, Fig. 2c) and that repeat tract breakage is promoting the increase in TEF promoter recombination. The genesis of the second break could be due to a nuclease targeting the CTG hairpin and if so, removal of these nucleases would result in increased viability in the $(CTG)_{70}$ strain post-DSB induction. We deleted structure specific endonucleases that have been previously shown to target DNA hairpins, such as Mlh1, Mus81 and Slx1[29–31] and found no increase in viability compared to wildtype (Supplementary Fig. 4d), suggesting either there is redundancy between nucleases or

that different nuclease is responsible for the breakage at the CTG repeat. Regardless, these results indicate that breakage occurs at the $(CTG)_{70}$ tract when it becomes single-stranded, resulting in alternate repair events that cause a loss of viability.

We considered whether the second break and alternative repair between promoters could help explain the increased ssDNA profile observed during resection across the $(CTG)_{70}$ template (Fig. 2b). BIR occurs via conservative DNA synthesis and has asynchronous replication of the leading and lagging strands[32]. If repair at the TEF promoter were occurring via BIR and synthesizing to the telomere end via D-loop bubble migration, then persistent, increased ssDNA would be observed at sites within the BIR tract (Supplementary Fig. 4e). Consistent with delayed gap filling due to BIR, increased, persistent ssDNA was detected at a location ~5 kb after the repeat tract in the $(CTG)_{70}$ template strain compared to the scrm$(CTG)_{70}$ control (Fig. 4f).

We also tested whether a possible reason for the increased BIR between the *HPH* and *NAT* loci was from CTG breakage events that occurred after the repeat was single-stranded but before resection exposes the U2 homology. Supportive of this hypothesis, DNA 2.3 kb before the U2 homology (i.e., ~23 kb from the HO-induced DSB) is minimally single-stranded at hour 4 when the break at the $(CTG)_{70}$ repeat tract occurs (Supplementary Fig. 4g). Interestingly, at this location, the resection is significantly delayed once the break at the repeat tract occurs, suggesting the possibility that the DNA substrate accessibility to exonucleases changed because of the ongoing BIR nearby. In summary, exposure of ssDNA by resection over the structure-forming CTG tract creates a highly fragile site, resulting in altered repair, large-scale deletions of neighboring genes, and cell death.

### Regulation of resection by Rad9 increases repair kinetics to rescue viability and repeat-induced contractions
We next wanted to determine what genetic factors could impact repair efficiency during gap filling of a $(CTG)_{70}$ template. Rad9 is an evolutionarily conserved DNA damage checkpoint protein that also has functions in restricting resection[7]. Loss of Rad9 results in increased recruitment of RPA, Rad51, and Rad52 on resected DNA around a DSB[33]. Intriguingly, deletion of *RAD9* almost completely rescued the loss in viability in the $(CTG)_{70}$ template strain, but had no impact on viability in the scrm$(CTG)_{70}$ strain (Fig. 5a). In addition, deletion of *RAD9* showed a significant reduction in the frequency of repeat contractions during gap filling across the CTG tract (Fig. 5b). It was previously established that resection speed in *rad9Δ* mutants is twice as fast as wildtype[9]. To test the role of Rad9 in resection in the presence of a structure-forming repeat, ssDNA levels were measured over time course, monitoring a site after the repeat tract (primers P5 & P6). The resection kinetics were altered in the *rad9Δ* mutant, showing maximal ssDNA 2–4 h earlier than in wildtype strains, and this was independent of the presence of the repeat tract (Fig. 5c, Supplementary Fig. 5a).

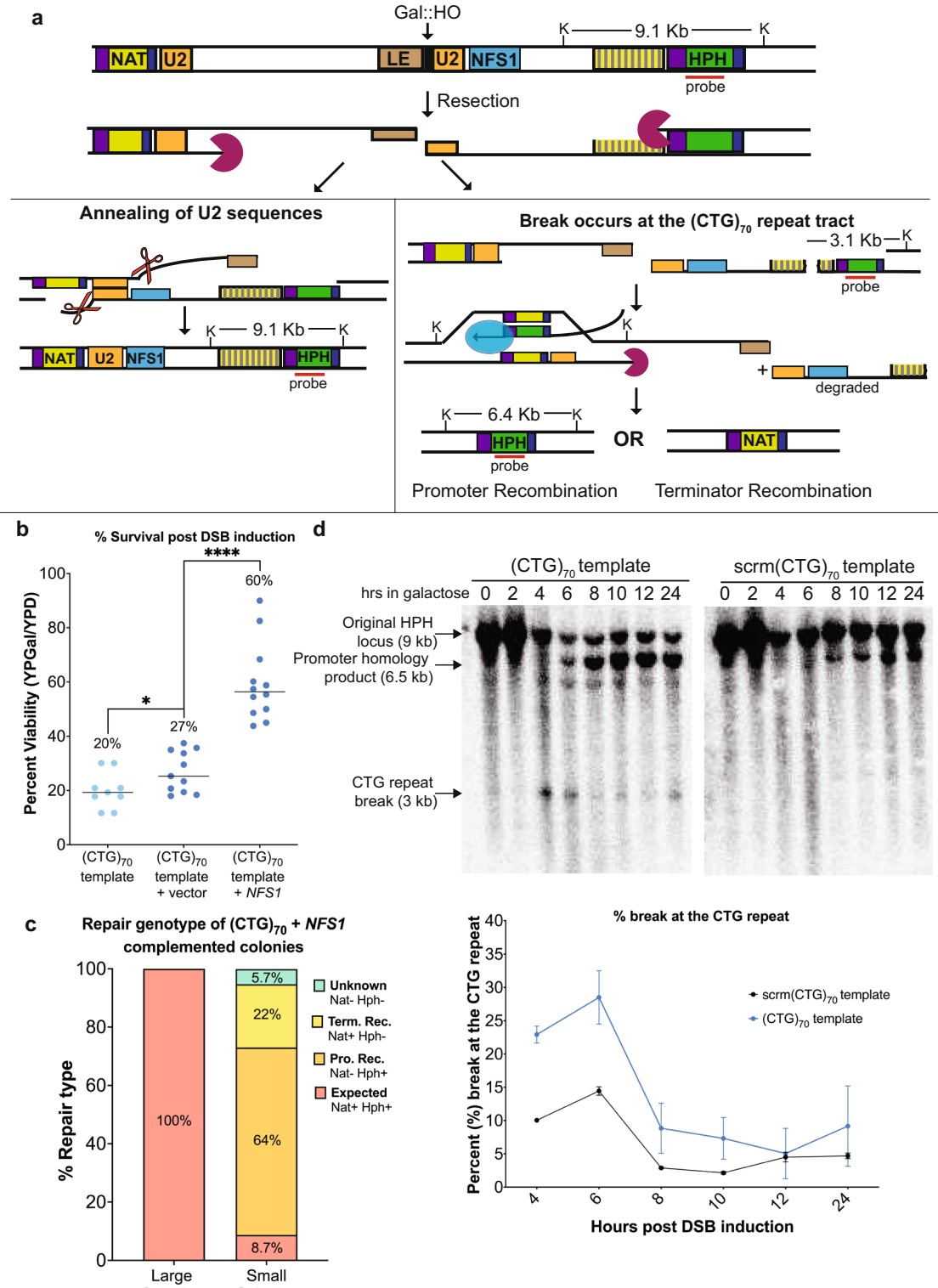

Unlike in the wildtype $(CTG)_{70}$ template strain, the $rad9\Delta$ mutant does not have persistent ssDNA at later time points (Fig. 5c). Consistent with faster resection and gap filling, there was a significantly earlier appearance of the U2 repair product in the $rad9\Delta$ mutant compared to wildtype (Fig. 5d, Supplementary Fig. 5b). Further the amount of U2 repair in the $rad9\Delta$ mutant was significantly increased compared to the wildtype $(CTG)_{70}$ template strain (Fig. 5d). The increased speed of resection, repair, and fill-in in the $rad9\Delta$ mutant compared to wildtype could explain the rescue in $(CTG)_{70}$ strain viability. Indeed, CTG repeat breakage could not be detected in the $rad9\Delta$ mutant when Southern

blots monitoring U2 repair were re-probed to the $HPH$ locus (Supplementary Fig. 5c), indicating that loss of Rad9 results in fewer breaks at the repeat locus. Consistent with the lack of breakage at the CTG repeat tract in the $rad9\Delta$ mutant, there is decreased levels of TEF promoter recombination compared to wildtype for the $(CTG)_{70}$ template strain (Supplementary Fig. 5d). To test whether faster recruitment of RPA and Rad51 could help protect ssDNA from hairpin formation and nucleolytic cleavage, we employed ChIP analysis over time post-DSB induction. Deletion of $RAD9$ resulted in earlier recruitment of RPA and Rad51 at the $(CTG)_{70}$ tract compared to the wildtype

**Fig. 4 | A second DNA break occurs at the $(CTG)_{70}$ repeat and results in decreased viability. a** The model of the two-break hypothesis in the $(CTG)_{70}$ template. Upon DSB induction, resection occurs on both sides of the break. *Left*, if repair occurs as expected, the two U2 regions of homology anneal, and error-prone gap filling occurs through the repeat tract leading to contractions. *Right*, if a second break occurs at the $(CTG)_{70}$ repeat tract during resection, then repair occurs via BIR using the TEF promoter or TEF terminator sequences as homologies which are present in nearby marker genes. Recovered alternatively repaired colonies are either NAT+ or HPH+. The genomic region between the two breaks spans chromosome III position 92418-104619. The ssDNA fragment is subject to exonucleolytic degradation and loss of the essential gene *NFS1*, as well as the *DCC1*, *BUD3*, *YCL012C*, *GBP2*, and *SGF29* genes. *HPH* probe and relevant *Kpn*I sites and expected sizes using the *HPH* probe are marked. The expected size of the *Kpn*I-digested fragment of the original *HPH* locus is 9.1 kb and recombination products between the TEF promoter (Promoter homology product) produces a band of ~6.4 kb. If a break occurs at the CTG repeat, a band ~3.1 kb is expected. **b** Percent viability of $(CTG)_{70}$ template ($n = 9$), $(CTG)_{70}$ template + vector ($n = 11$) and $(CTG)_{70}$ template +*NFS1* ($n = 12$) are shown where n represents assays from biologically independent experiments. Statistical significance was determined by Student's $t$ test (two-tailed, unpaired). Comparison of $(CTG)_{70}$ no vector to $(CTG)_{70}$ + vector is $p = 0.04$ and $(CTG)_{70}$ + vector to $(CTG)_{70}$ + *NFS1* is $p > 0.0001$. **c** Genetic typing of large ($n = 69$) and small ($n = 115$) repaired colonies in the $(CTG)_{70}$ template strain that was complemented with *NFS1*. **d** CTG repeat breakage (%) on Southern blots after DSB induction. Kinetic Southern blots of KpnI digested DNA were stripped and probed with a fragment to the *HPH* locus. Representative Southern shown; number of replicates: scrm$(CTG)_{70}$ ($n = 2$) and $(CTG)_{70}$ ($n = 4$) where n represents biologically independent time courses. For (**b**–**d**) Source data are provided as a Source Data file.

(Fig. 5e & f, Supplementary Fig. 5e & f; see hours 2–4), which is consistent with the earlier peak of ssDNA accumulation at 2 h (Fig. 5c). Correspondingly, RPA and Rad51 are removed more quickly in the *rad9Δ* mutant, with reduced levels by 6 h, whereas they are maximal at 6 h in wildtype cells. Taken together, this data suggests that the rescue in $(CTG)_{70}$ contractions and viability in the *rad9Δ* mutant is due to faster repair kinetics which results in less opportunity for CTG hairpin formation and tract breakage.

### Rad51 protects the single-stranded $(CTG)_{70}$ repeat from fragility and contractions

Due to the extensive amount of ssDNA produced in this assay and its importance in determining repair outcome, we tested the role of Rad51, which binds ssDNA to promote strand exchange. Rad51 is not required for repair via SSA[13], but it has been shown that the parent assay system can repair via either SSA or Rad51-dependent BIR with similar kinetics[19]. Indeed, Rad51 is recruited to the ssDNA during resection in both the scrm$(CTG)_{70}$ and $(CTG)_{70}$ template strains (Supplementary Fig. 2a–d). Deletion of *RAD51* in the $(CTG)_{70}$ template strain resulted in a significant decrease in viability (Fig. 6a). Deletion of *RAD51* in the scrm$(CTG)_{70}$ template also had a decrease in viability, though less dramatic (Fig. 6a). Lack of Rad51 could result in less protection of the repeat on the single-stranded template leading to secondary structures that are targets of nucleolytic cleavage or contraction intermediates (Fig. 4a). Consistent with this idea, loss of Rad51 increased the frequency of CTG repeat tract contractions during gap filling to 98% (Fig. 6b). Deletion of *RAD51* in a *rad9Δ* mutant resulted in an even lower level of viability in both the scrm$(CTG)_{70}$ and $(CTG)_{70}$ template strains compared to the *rad51Δ* single mutant, eliminating the rescue observed in the *rad9Δ* mutant (Fig. 6a). Similarly, CTG contractions were still high in the *rad9Δ rad51Δ* double mutant with no rescue (Fig. 6b). Therefore, Rad51 functions upstream of Rad9 in gap repair. We conclude that in the absence of Rad51 to protect the resected DNA and/or facilitate BIR-repair, faster resection in the *rad9Δ* mutant cannot rescue repeat contractions due to template hairpin formation or inviability due to breaks.

To better determine whether the addition of a repeat tract changes repair kinetics in the *rad51Δ* mutant, we followed the time course of the U2 repair reaction by Southern blot (Fig. 6c). Consistent with previous work[19], the U2 repair product was delayed in the *rad51Δ* mutant strains (Fig. 6c), first appearing around 8 h and then increasing slowly from 10 to 24 h (Supplementary Fig. 6a). Breaks that occur at the CTG repeat tract appear when the DNA first becomes single-stranded at hour 6 and persist through hour 12 in the *rad51Δ* mutant (Supplementary Fig. 6b, c). Alternative recombination between the TEF promoters in the *rad51Δ* mutant is also reduced compared to wildtype in the $(CTG)_{70}$ template strain (Fig. 6d, Supplementary Fig. 6b) suggesting that Rad51-dependent repair is driving promoter recombination when there is a break at the CTG repeat.

Given that Rad51 seems to play a protective role at the CTG repeat tract, we measured the kinetics of generating ssDNA after the repeat locus in the *rad51Δ* mutant. Interestingly, *rad51Δ* mutants had a 2–4 h delay in resection through the repeat locus in the $(CTG)_{70}$ and scrm$(CTG)_{70}$ template strains (Fig. 6e, Supplementary Fig. 6d). Confirming the delayed resection phenotype in the *rad51Δ* mutant, delayed enrichment of RPA was also observed (Fig. 6f, Supplementary Fig. 6e). The delay in ssDNA accumulation is consistent with the delay in U2 repair in these strains. Strikingly, the region after the $(CTG)_{70}$ template is maximally single-stranded through the 12-h timepoint in the *rad51Δ* mutant, much later than the 6-h peak in wildtype cells (Fig. 6e). There is also increased ssDNA in the *rad51Δ* mutant in the scrm$(CTG)_{70}$ strain though it does not persist to the same degree (Supplementary Fig. 6d). Thus, a deficiency in Rad51 binding causes ssDNA persistence, which is exacerbated when a structure-forming CTG repeat is on the exposed single-stranded template strand.

The loss of the competing BIR pathway in the *rad51Δ* mutant could explain the delayed appearance of the U2 repair product, the decrease in the TEF promoter product, the persistent ssDNA and the additional loss of viability in this mutant. To confirm this reasoning, genetic complementation with *NFS1* was used in the *rad51Δ* mutant with the rationale that if the small colonies seen in the $(CTG)_{70}$ template strain were mostly products of BIR a decrease in the small colony phenotype and no rescue in viability would be seen in the *rad51Δ* mutant compared to wildtype. Supportive of our hypothesis, a decrease in the number of small colonies in the *rad51Δ* mutant complemented with *NFS1* was observed (Supplementary Fig. 6f). Quantitatively, there is a significant rescue in the viability in the *rad51Δ* mutant $(CTG)_{70}$ template strain complemented with *NFS1*, however, the percent viability is much lower than the viability of the wildtype complemented with *NFS1*, only reaching 25% (Supplementary Fig. 6g). Together, this suggests that while cells can heal the CTG break without Rad51, a Rad51-dependent process like BIR is preferred. Given the decreased viability and reduced healing in the *rad51Δ* mutant, we asked whether it has a defect in the resumption of cell division post-DSB induction in the $(CTG)_{70}$ template strain. Following *rad51Δ* mutant cells over 24 h showed that only 14% of cells were able to complete more than 5 divisions, suggesting some *rad51Δ* mutant cells can eventually either heal or adapt, but many are permanently arrested (Supplementary Fig. 6h). The surviving *rad51Δ* mutant cells can extinguish the DNA damage checkpoint by 24 h post-DSB induction (Supplementary Fig. 6i), though this is delayed compared to the 12 h observed in wildtype cells (Supplementary Fig. 1d). Therefore, a small proportion of *rad51Δ* mutant cells with a CTG break can eventually repair the break by SSA, extinguish the DNA damage checkpoint and resume division, but the majority are permanently arrested and die.

Taken together, these data reveal a dual role for Rad51, first in protecting repeat tracts from forming DNA structures during resection which prevents contractions and fragility, and second by providing BIR as a repair pathway choice. Repair via BIR may be especially important

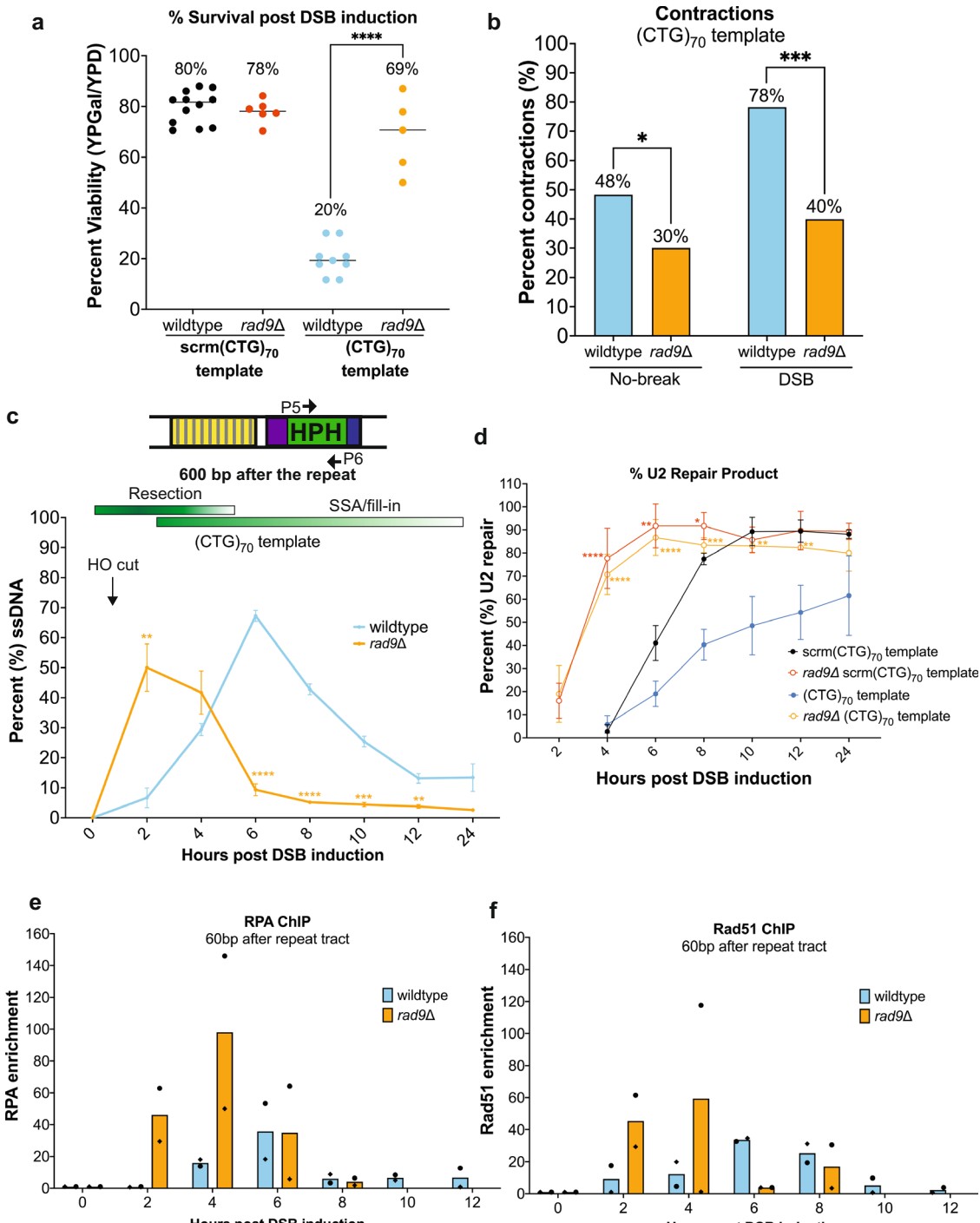

**Fig. 5 | Deletion of *RAD9* rescues the decreased viability and gap fill-in mediated (CTG)$_{70}$ contractions. a** For the scrm(CTG)$_{70}$ template strain, percent viability of the *rad9Δ*mutant (*n* = 6) is unchanged compared to wildtype (*n* = 12). For the (CTG)$_{70}$ template strain, percent viability of the *rad9Δ* mutant (*n* = 5) is significantly increased compared to wildtype (*n* = 9) (*p* = 0.0002). Each *n* value represents assays from biologically independent experiments. Statistical significance determined using an unpaired, two-tailed Student's *t* test. **b** In the (CTG)$_{70}$ template strain, the contraction frequency in the no-break condition (*n* = 120) decreased in the *rad9Δ* mutant (*n* = 163), *p* = 0.04. In the DSB condition, the contraction frequency decreased more significantly in the *rad9Δ* mutant (*n* = 163), *p* = 0.001 compared to wildtype (*n* = 119). Each n value represents a PCR of an independent colony, statistical analysis by Fisher's exact test. **c** Percent ssDNA 600 bp after the repeat locus was determined after DSB induction as in Fig. 2b; wildtype *n* = 4, *rad9Δ n* = 3. Graph shows mean ± SD where *n* represents biologically independent time courses.

Statistical significance determined unpaired Student's *t* test using a two-stage step-up with a false-discovery rate of 1% (Benjamini, Krieger, and Yekutieli). **d** U2 repair measurement (%) on Southern blots after DSB induction. Statistical significance determined unpaired Student's *t* test using a two-stage step-up with a false-discovery rate of 1% (Benjamini, Krieger, and Yekutieli) where *rad9Δ* scrm (CTG)$_{70}$ (*n* = 3) was compared to scrm(CTG)$_{70}$ (*n* = 3) and *rad9Δ* (CTG)$_{70}$ (*n* = 3) was compared to (CTG)$_{70}$ (*n* = 4). Each *n* represents biologically independent time courses. Enrichment of (**e**) RPA and (**f**) Rad51 60 bp after the (CTG)$_{70}$ repeat tract occurs earlier in the *rad9Δ* mutant following DSB induction compared to wildtype. Independent biological replicates for wildtype (*n* = 2) and *rad9Δ* (*n* = 2). Enrichment adjacent to the (CTG)$_{70}$ repeat was determined using primers P3 & P4 and calculated using absolute quantity and normalized to *ACT1*. Bars on graph depict the mean; • and ◆ each indicate one experimental replicate. For (**a**–**f**) source data are provided as a Source Data file.

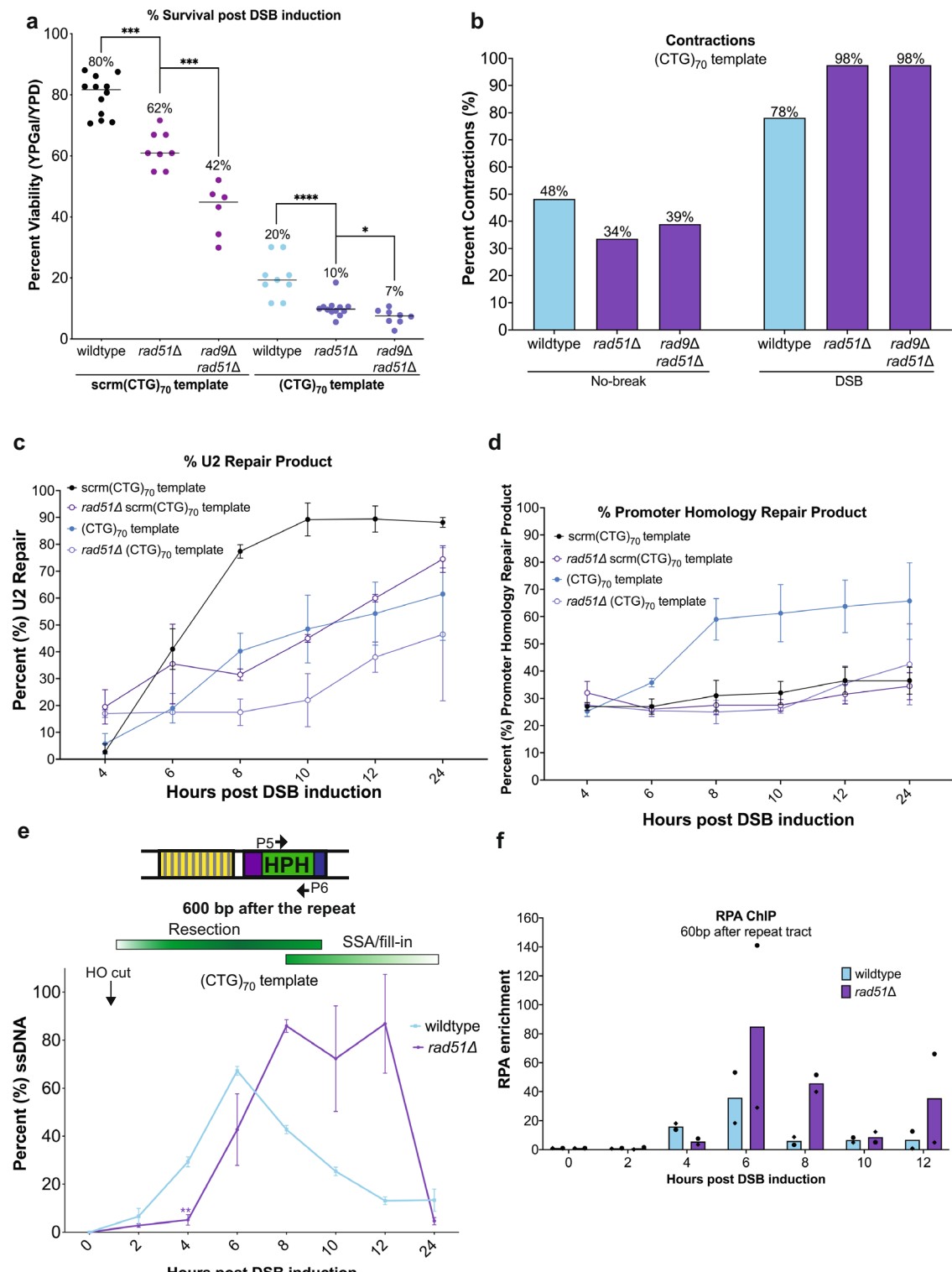

## Discussion

Previous work had shown that CAG repeat expansions and contractions can occur during HR repair, but it was not clear which steps of HR were involved. In this study, we developed an assay system to test the stability of expanded CAG/CTG trinucleotide repeats specifically during the gap filling step of HR. Our data show that both resection and subsequent DNA polymerization during gap repair are mutagenic when there are breaks due to a fragile sequence within a single-stranded gap, as other repair pathways may not be readily available.

steps of HR when a structure-forming CAG/CTG tract is present. The identity of the repeat tract in relation to the resected strand results in very different phenotypes. The $(CAG)_{70}$ template, with CAG on the ssDNA template strand and CTG on the 5' recessed end, has resection defects and is prone to gap fill-in mediated expansions. When the repair template strand has a $(CTG)_{70}$ repeat there are two possible repair outcomes (Fig. 4a). First, if gap filling occurs as expected this frequently results in large-scale contraction of the CTG repeat tract. The surprising second outcome is a repeat tract-dependent single-strand break that occurs after resection exposes the hairpin-forming

**Fig. 6 | Deletion of *RAD51* impairs repair and results in increased contractions during gap filling. a** For the scrm(CTG)$_{70}$ template strain: viability (%) of the *rad51Δ* mutant (*n* = 8) is significantly decreased compared to wildtype (*n* = 12) (*p* = 0.002) and the *rad9Δ rad51Δ* mutant (*n* = 6) is significantly decreased compared to the *rad51Δ* mutant (*p* = 0.0002). For the (CTG)$_{70}$ template strain: viability (%) of the *rad51Δ* mutant (*n* = 12) is significantly decreased compared to wildtype (*n* = 9) (*p* = 0.0002) and the *rad9Δ rad51Δ* mutant (*n* = 8) is significantly decreased compared to the *rad51Δ* mutant (*p* = 0.04). Each *n* value represents assays from biologically independent experiments. Statistical significance was determined using an unpaired, two-tailed Student's *t* test. **b** No break condition: *rad51Δ* (*n* = 119) and *rad9Δ rad51Δ* (*n* = 96) mutants had decreased contractions compared to wildtype (*n* = 120; *p* = 0.15; *p* = 0.38 respectively). DSB condition: *rad51Δ* (*n* = 119) and *rad9Δ rad51Δ* (*n* = 120) mutants had increased contractions compared to wildtype (*n* = 119; *p* = 0.26 for both). Each *n* value represents a PCR of an independent colony, statistical analysis by Fisher's exact test. **c** U2 repair measurement (%) on Southern blots after DSB induction. Number of replicates: scrm(CTG)$_{70}$ (*n* = 3), *rad51Δ* scrm (CTG)$_{70}$ (*n* = 2), (CTG)$_{70}$ (*n* = 4), and *rad51Δ* (CTG)$_{70}$ (*n* = 2) where each *n* represents biologically independent time courses. Graph shows mean ± SD. **d** Promoter homology repair measurement (%) on Southern blots after DSB induction. Number of replicates: scrm(CTG)$_{70}$ (*n* = 3), *rad51Δ* scrm(CTG)$_{70}$ (*n* = 2), (CTG)$_{70}$ (*n* = 4), and *rad51Δ* (CTG)$_{70}$ (*n* = 2) where each n represents biologically independent time courses. Graph shows mean ± SD. **e** Resection and fill-in kinetics for the (CTG)$_{70}$ template in wildtype (*n* = 4) and *rad51Δ* (*n* = 2) strains 600 bp after the repeat locus after DSB induction where each *n* represents biologically independent time courses. **f** Enrichment of RPA 60 bp after the (CTG)$_{70}$ repeat tract in wildtype and *rad51Δ* mutants following DSB induction. Independent biological replicates for wildtype (*n* = 2) and *rad51Δ* (*n* = 2) where each n represents biologically independent time courses. Enrichment was determined using P3 & P4 and calculated using absolute quantity and normalized to *ACT1*. Bars on graph depict the mean; • and ♦ each indicate one experimental replicate. For (**a**–**f**) source data are provided as a Source Data file.

repeat, leading to altered repair and cell death. HR is highly conserved between yeast and mammals and our findings provide insight on how the region surrounding a DSB and its structure-forming potential plays a key role in repair fidelity and outcome.

In the situation where the gap repair template strand was an expanded (CAG)$_n$ tract (CAG template) and the more stable CTG structure could form on the 5′ resected end, there were decreased levels of resection beyond the repeat and an increase in expansions. Supportive of the possibility that non-B form structures can be a barrier to resection, it has been shown that resection is impeded, and repair choice is altered in the presence of a stabilized G-quadruplex on the resected strand[34]. Because resection of this CTG-containing strand to the distance of the TNR is not imperative for repair, viability and repair kinetics were unaltered. However, repeat stability was altered so that gap fill-in outcomes favored repeat expansion. The decreased resection coupled with the increase in expansions provides a simple model for repeat expansion due to incorporation of an unprocessed structure on the 5′ flap during the final ligation step of gap filling (Fig. 3f). What is surprising about these expansions is how relatively small they are, as most of them are an addition of only 1 or 2 repeat units (3-6 bp). These small expansions stand in contrast to large-scale CAG expansions that were shown to occur during a BIR-like process dependent on Rad52 and Pol32[35,36]. It suggests that the hairpins formed on the resected strand are relatively small. It is also possible that the DNA structure impairing resection is resolved, and that the expansion event is due to polymerase slippage, or that the two processes are linked (Fig. 3f). These small-scale expansions could be like those that occur during gap repair in non-dividing human cells, such as neurons, that undergo stepwise somatic expansions which advance disease onset in TNR expansion disorders such as Huntington's disease[6]. Modeling of the expansion bias observed in humans using a large sample of blood DNA samples collected from Myotonic Dystrophy type 1 (DM1) patients predicted that hundreds of small expansion and contraction events accumulate over the lifespan of a hematopoietic stem cell, and the rate was consistent with occurrence due to DNA damage and repair rather than replication[37]. Thus, our data provide a mechanism for small TNR expansion events that can result in worsening of disease phenotypes in somatic cells.

In the situation where the gap repair template strand was an expanded (CTG)$_n$ tract (CTG template), contractions predominated and there was a loss in viability. In cells that repaired as expected (using the U2 homology), the CTG repeat likely forms several hairpins which are bypassed during gap filling leading to a variety of large-scale contractions (Figs. 3e, 7a). Alternatively, DNA breaks in the single-stranded CTG repeat tract could result in contraction via out of register alignment (Supplementary Fig. 3e). This second model for repeat contractions is similar to one proposed for Cas9-induced breaks at an expanded CAG/CTG repeat which repair via end joining or SSA[24].

However, there are differences, as the Cas9-induced break is directly targeting the double-stranded repeat whereas in our assay the repeat tract becomes single-stranded due to resection first and then is converted to a break. Nonetheless, when designing strategies for nick-induced repeat contractions, it should be considered that the template strand composition most likely to result in gap fill-in contraction events may also result in increased breakage and cell death.

The breakage of the ssDNA during resection and gap repair of the (CTG)$_{70}$ template resulted in a frequent large, lethal deletion. The nuclease responsible for the genesis of the repeat-dependent break is still unknown, but appears to not be Mlh1, Mus81, or Slx1 acting individually. One nuclease that could be targeting the CTG repeat is Mre11 as it is required for processing hairpin-capped ends[38] and creates breaks at inverted repeats that form hairpins on ssDNA tracts that occur during lagging strand replication[31]. However, Mre11 is required for the initiation of resection, and loss of the MRX complex or Mre11's nuclease activity result in significantly decreased viability in this strain[10,39], preventing a test to determine whether Mre11 is the nuclease responsible for the break at the CTG repeat tract.

In this assay system, the large gap (up to 25 kb) creates a long stretch of ssDNA and a significant need for ssDNA protection during repair. Our data reveal that this situation is particularly dangerous for the cell as the ssDNA is prone to breakage, and this leads to large deletions and loss of vital genetic material (Fig. 7a). The significant difference between having the CAG or CTG strand on the exposed template indicates that structure-forming potential is a key determinant of whether a gapped ssDNA region results in chromosome fragility. Similar events could be behind large deletions that often occur at fragile sites in the human genome[40].

One of the more surprising findings was that repeat instability and fragility of the (CTG)$_{70}$ template was reduced in the absence of Rad9, the *S. cerevisiae* ortholog of mammalian 53BP1. Therefore, Rad9's normal function in slowing down resection can have a detrimental effect during repair of gaps containing structure-forming repeats, suggesting a model where faster recruitment of RPA and Rad51 help guard against hairpin formation on the exposed single-stranded template. RPA is a trimeric complex that normally coats ~25 nt of ssDNA[41] while Rad51 is much smaller and coats ~3 bp of ssDNA[42]. Since we see RPA and Rad51 accumulate as the DNA is resected, one possibility is that in wildtype cells a brief lag between resection and RPA or Rad51 loading normally occurs[43] and is sufficient to allow formation of small hairpins that form as resection proceeds (Fig. 7a). Loss of Rad9 may eliminate the lag between resection and RPA or Rad51 loading, resulting in fewer DNA hairpins and a more stable repeat tract (Fig. 7b). Rad9 may also influence loading of other repair factors that play a role in repeat stability. For example, Rad9 limits recruitment of the helicases Mph1 and Sgs1 to resected DNA[33]. Since non-B form DNA structures that form due to TNRs are unwound by helicases to prevent

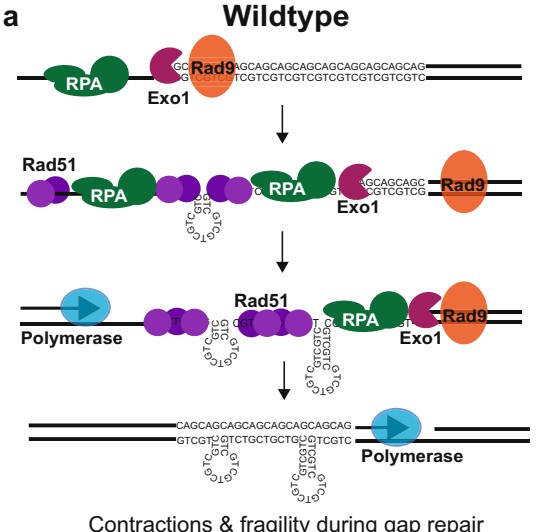

## Wildtype

**a**

Contractions & fragility during gap repair

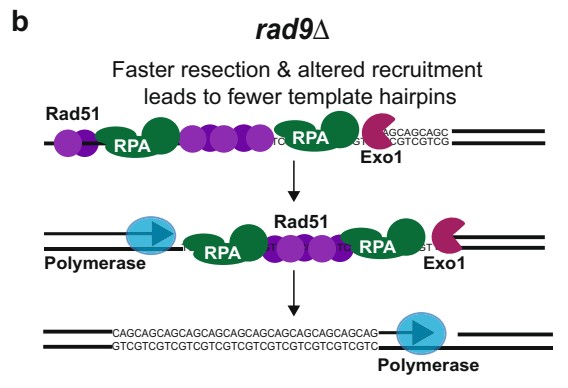

**b**  *rad9Δ*

Faster resection & altered recruitment leads to fewer template hairpins

Less contractions & fragility during gap repair

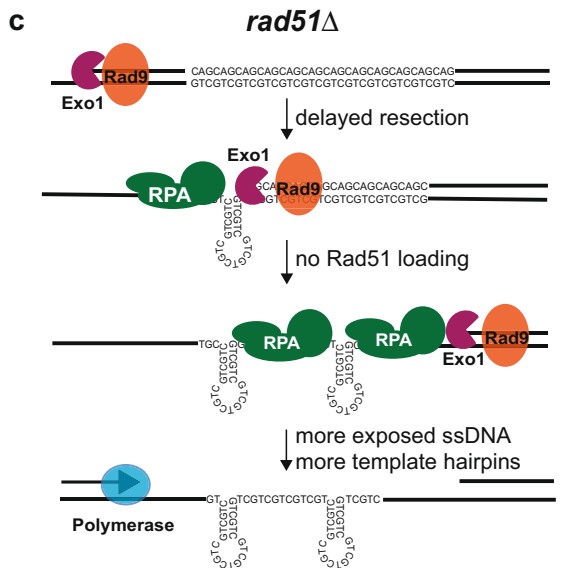

**c**  *rad51Δ*

delayed resection

no Rad51 loading

more exposed ssDNA
more template hairpins

More contractions, fragility, impaired repair

**Fig. 7 | A model for repeat instability during resection and gap filling at a CTG repeat. a** In wildtype cells, as resection occurs, Rad9 controls the recruitment of RPA and Rad51 to ssDNA. Delayed binding of RPA or Rad51 results in hairpin formation. Contractions are a result of hairpin bypass during polymerase mediated gap filling (shown) or by SSA/end-joining events within the broken CTG repeat region (Supplementary Fig. 3g). In addition, exposed ssDNA results in increased breakage and alternative repair. **b** In *rad9Δ* mutants, there no delayed recruitment of RPA and Rad51 to the freshly resected ssDNA. This prevents hairpin formation and breakage at the repeat tract resulting in increased viability and decreased contractions. **c** In *rad51Δ*, initiation of resection is delayed. Absence of Rad51 and slowed gap filling both allow for increased DNA structure formation and contractions. Breaks that occur at the CTG repeat tract cannot repair via BIR leading to increased cell death.

repair mediated contractions and inviability when $(CTG)_{70}$ was the fill-in template. The increase in contractions suggests that Rad51 may have a protective role in gap repair by binding ssDNA and preventing secondary structure formation (Fig. 7c). Interestingly, *rad51Δ* mutants have a slower rate of 5' to 3' resection and this is mirrored by delayed RPA recruitment, even in strains without a repeat tract. However once resection starts, *rad51Δ* mutants accumulate more ssDNA that persists for longer compared to wildtype. Our data suggest that if a second break occurs at the CTG repeat, BIR using the MX homologies becomes the preferred mechanism for repair (Fig. 4a, Supplementary Fig. 4f). Elimination of the BIR pathway by deletion of *RAD51* leaves cells with a break within an ssDNA region that must repair via SSA using homologous sequences that aren't yet resected, and this is a major cause for the delayed repair and loss in viability in the *rad51Δ* mutant (Fig. 7c).

Gap filling is not a process unique to HR and our findings could also apply to other repair pathways that involve a gap filling step. Cells that repair DSBs using MMEJ exhibit increased mutation levels, presumably due to extensive resection of DSBs and mutagenic gap filling[46]. Mismatch, base excision, and nucleotide excision repair pathways all involve gap filling post-removal of a lesion, and repeat instability in somatic tissues affected in TNR diseases has been attributed to these pathways[23]. Our findings could also help explain how TNR expansions accrue in non-dividing cells during aging[23]: as there are increased DSBs in neuronal tissues with age[47], repair of neuronal DSBs near sites of TNRs could lead to gap filling mediated instability, or alternative repair that could result in cell death.

Our results indicate that gap repair is mutagenic in the context of a structure-forming DNA sequence. This mechanism could be particularly relevant to cancer cells which have lots of gapped DNA[48–50]. It should also be considered when designing target sites for gene editing: if a repetitive, structure-forming sequence is near a DSB target locus, it is possible that a secondary break and alternative repair could result. Our findings indicate that the success and accuracy of repair is influenced by the sequence context where repair is occurring and illustrate the danger of exposing ssDNA within repetitive sequences during gap repair.

## Methods

### Yeast strains

All strains are derivatives of YMV80[13] which was derived from S288C. $(CAG)_{70}$, $(CTG)_{70}$ and scrm$(CTG)_{70}$ repeat tracts were integrated at the *ILV6* locus on chromosome V. Proper integration of the repeat tract was confirmed via Southern blotting and expected tract length was confirmed via PCR. Gene deletions were generated by one step gene replacement with a marker gene. All strains are listed in Supplementary Table 1.

### Viability assay

Tract length was confirmed via colony PCR using primers listed in Supplementary Table 1. Size was determined by either electrophoretic

fragility and instability[23], another possibility is that the Rad9 mediated rescue is due to improved unwinding of the CTG hairpin. 53BP1 and Rad9 share many of the same functions relating to resection and repair pathway choice[44,45], thus 53BP1 might also affect how structure-forming DNA is processed at gaps and broken ends in human cells.

We identified novel roles for Rad51 in long-range resection and in protection of repetitive ssDNA. Loss of *RAD51* resulted in increased gap

analysis using a fragment analyzer or 2% metaphor agarose. Colonies with confirmed tract length were inoculated into 2 mL YP + Lactate (pH 5.5) and grown for 2–3 divisions (16–18 h). Cultures were appropriately diluted and plated on YPD and YP + 2% Galactose (YPGal) in duplicate. Plates were incubated at 30 °C for 2–3 days. Colonies were counted and percent viability was obtained by dividing the number of colonies of galactose by the number of colonies on glucose multiplied by 100. See source data for individual assay values.

## Statistics

For all instability assays, significance was determined using Fisher's exact test. For viability, significance was determined using a two-tailed unpaired Student's *t* test. For resection assays and Southern repair quantification, an unpaired Student's *t* test was used which compared the SD of each timepoint using a two-stage step-up with a false-discovery rate of 1% (Benjamini, Krieger, and Yekutieli). For all assays, standard significance denotations are used: *$P > 0.05$, **$P > 0.01$, ***$P > 0.001$, ****$P > 0.0001$.

## DSB repair Southern blotting

Time course collection was adapted from[13]. Colonies of the correct tract length were inoculated into 10 mL YP + Lactate pH 5.5 for 24 h. Cultures were diluted into 400 mL YP + Lactate pH 5.5 and grown for approximately 12–14 h. Prior to galactose addition, cells went through approximately 6–7 cell divisions and the final cell number prior to addition of galactose was approximately $7 \times 10^6$ cells per milliliter. Galactose was added to a final concentration of 2% and samples were taken at the indicated times. DNA was prepped via phenol-chloroform extraction and normalized via Qubit (dsDNA BR Assay Kit, Cat# Q32853; Invitrogen). DNA was digested with KpnI, separated on a 0.8% agarose gel, blotted, and probed with a fragment within the *LEU2* gene or *HPH* marker. Bands were visualized with a Typhoon phosphorimager (GE Biosciences). Blots were quantified using ImageJ. For Southerns probed with the LEU2 probe, determination of percent U2 repair was done by measuring the amount of signal of the donor U2 and the U2 repaired band and percent repair was determined using the following equation (U2 repaired band/(U2 repaired band + U2 donor band)*100). For Southerns probed with the HPH probe, the total signal of the original HPH locus, the promoter homology product (PHP) and break at the CTG repeat were measured. PHP percentage was determined by ((PHP/PHP + break at repeat+original HPH locus)*100). Quantification of the break at the repeat was determined by ((break at repeat/PHP + break at repeat + original HPH locus)*100).

## Resection/gap fill-in assay

Adapted from ref. [16]. DNA from the kinetic time courses was normalized using a Qubit. For each digest, 150 ng of DNA was digested with EcoRI or mock treatment overnight at 37 °C such that the final concentration was 2 ng/ul. Each quantitative PCR (qPCR) reaction had 20 ng DNA and was run in duplicate. qPCR was run using a QuantStudio 6 Real Time PCR machine (Applied Biosystems). Determination of percent resected was done using %ssDNA = [100/ [(1 + 2^{ΔΔCt})/2]/*f*] where $\Delta\Delta Ct = \Delta Ct_{digested} - \Delta Ct_{mock}$ and *f* is HO cutting efficiency. HO cutting efficiency (*f*) was obtained using densitometric analysis using Imagequant. HO cutting efficiency was calculated as *f* = Cut value/ (Cut value+Uncut value). Each resection assay has a paired Kinetic southern blot to monitor repair efficiency and was used to determine cutting efficiency. Individual values of each experimental replicate and primer set tested are listed in source data.

## CAG/CTG repeat size analysis

For tract length analysis, resulting colonies from the viability analysis were used. Colony PCR was performed on 22-24 daughter colonies from both the paired YPD and YPGal plates using primers that span the repeat tract (Fig. 1A). To eliminate variation, PCR and subsequent electrophoretic separation of repeat amplicons of daughter colonies from the YPD and YPGal conditions were done at the same time. For primers used see Supplementary Table 1. For size analysis, PCR amplicons were sized on a fragment analyzer (Model# 5200; Agilent) using 600mer DNA separation gel (Cat# NDF-915-0275; Agilent) compared to a 1 bp & 6000 bp marker (Cat# FA-MRK915F-0003; Agilent) and a 100 bp DNA plus ladder (Cat# FS-SLR915-0001; Agilent). For the *rad9Δ* and *rad51Δ* mutants, PCR amplicons were separated using standard gel electrophoresis on 2% metaphor agarose and sized in comparison to a 100 bp Hyperladder (Cat# BIO-33056; Bioline).

In order to determine the repeat size distributions of the no-break and break conditions, tract lengths (in bp) determined by the fragment analyzer were plotted versus the number of times the fragment analyzer made the same size determination. The median tract length in the no-break condition was mathematically determined. If the median was between two whole numbers, both whole numbers were designated as the median to determine expansion and contraction cut offs. The PCR amplicon (Fig. 1a, Primers P1 & P2) encompasses 210 bp of the repeat tract and 150 bp of non-repetitive sequence. Due to the large spread of contractions, tract lengths below 300 bp were not used in the determination of the median in the (CTG)_{70} template. Expansions or contractions were any size called ±3 bp (one repeat unit) from the calculated median. Statistical significance was determined using Fisher's exact test. See source data to see the number of times each size was called by the fragment analyzer and for contraction and expansion frequencies for all strains listed.

## RPA and Rad51 chromatin immunoprecipitation

Time course was as described for the kinetic Southern blots. Samples for RPA and Rad51 ChIP were taken simultaneously from the same cultures. Cultures were crosslinked in 1% formaldehyde for 20 min and quenched with glycine (0.125 M final concentration) for 5 min. Immunoprecipitation was performed by incubating normalized samples with Protein G Dynabeads (Thermo Fisher, 10004D) pre-conjugated with α-RPA (concentration of antibody used: 2 µG; Agrisera AS07 214) or α-Rad51 (concentration of antibody used: 3 µG; PA5-34905, Thermo Fisher Scientific) for 2 h at 4 °C. Whole chromatin input and immuno-precipitated samples were subject to qPCR using primers 60 bp and 600 bp after the repeat locus (see Supplementary Table 1) or *ACT1* as a control. Enrichment was determined using absolute quantity and normalized to *t* = 0. See source data for individual values of each experimental replicate and statistical analysis.

## Reporting summary

Further information on research design is available in the Nature Portfolio Reporting Summary linked to this article.

## Data availability

The data supporting the findings in this study are available within the paper and its supplementary information files and are available from the corresponding authors upon request. Source data are provided with this paper.

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

## Acknowledgements

The authors want to thank Marjorie de la Rosa Mejia and James Patti for strain construction and experimental assistance. Research reported in this publication was supported by an American Cancer Society–Ellison Foundation Postdoctoral Fellowship PF-18-125-10-DMC to E.J.P. and the National Institute of General Medical Sciences of the National Institutes of Health under Award Number R01GM122880 to C.H.F., R35 GM144215 to C.H.F., P01GM105473 to J.E.H. and C.H.F., and R35 GM127029 to J.E.H. The content is solely the responsibility of the authors and does not necessarily represent the official views of the National Institutes of Health.

## Author contributions

Experimental Design: E.J.P. and C.H.F.; Data accumulation: E.J.P. and I.D.P.; Writing: E.J.P., J.E.H., and C.H.F.

## Competing interests

The authors declare no competing interests.
