## [Peer Review File · Nature Communications]

Structure-forming CAG/CTG repeats interfere with gap repair to cause repeat expansions and chromosome breaksREVIEWER COMMENTS

Reviewer #1 (Remarks to the Author):

Manuscript # NCOMMS-22-11274

Trinucleotide repeat (TNR) instability is associated with human neurodegenerative diseases and cancer and can occur during DNA replication and repair. HR is one of the major mechanisms that underlie large repeat instability. However, the precise mechanism by which HR mediates repeat instability remains unclear and needs to be elucidated. One of the critical steps in HR is strand resection, which can result in a large single-strand DNA gap. The process may allow the generation of secondary structures in the repeat tracts leading to repeat instability during double-strand break repair. In this study, the authors developed a system to detect the resection and ssDNA gap-filling on CAG or CTG repeats. Using the system, they found that when CTG repeats served as the single-stranded template, the repair of the gap promoted repeat deletions. In contrast, the repair resulted in repeat expansion when CTG repeats on the resected strand. They further demonstrated that Rad9 prevented resection and slowed down the gap-filling synthesis, and Rad51 inhibited repeat deletion. The results suggest that different repeat DNA sequences in the single-strand DNA gap govern the repair rate and repeat instability. The study provided new evidence to support a mechanism underlying trinucleotide repeat instability during HR through gap-filling synthesis. The conclusions of the study are supported by the comprehensive experimental data. However, some major and minor issues shown below need to be addressed.

Major issues

1. In "Abstract," it is unclear why it is important to make the discoveries reported in this study and how Rad9 and Rad51 coordinated with DNA sequences to determine repeat deletion or expansion. In fact, in "Introduction," the authors gave an excellent introduction on the research gap and provided the justifications for their study. The same points need to be also presented in the "Abstract."
2. In Figure 2b, it is surprising that the scrm (CTG)₇₀ template exhibited a higher percentage of ssDNA than (CAG)₇₀. Figure 2c showed that the scrm(CTG)₇₀ template exhibited the highest percentage of ssDNA. Why did this happen? Does this mean the gap-filling synthesis on the region upstream of the repeats was facilitated by the repeats because it had less ssDNA? Also, in lines 165-166, the conclusion is that a CTG hairpin on the 5'-recessed strand prevents resection. Was the conclusion drawn based on the results of Figure 2c? There is no clear discussion about this. A justification and rationale need to be provided here.
3. To provide direct evidence to support the roles of gap-filling synthesis by DNA polymerases *in vivo*, it is critical to test the effects of gap-filling synthesis on the accumulation of ssDNA using a DNA polymerase deficient strain.
4. It is also critical to show ssDNA accumulation on the (CAG)₇₀, (CTG)₇₀, and scrm(CTG)₇₀ directly rather than just show them on the upstream and downstream strands of the repeat tracts. This can be done using the PCR primers annealed at the random DNA sequences at both the 5'- and 3'-flanking regions of the repeats. The P1 and P2 in Figure 1 may be used as the primers for the experiments.
5. In SFigure 2, it is not surprising that no significant difference was detected in the enrichment of RPA and Rad51 on the 600 bp fragment that is located at downstream of the repeats. Because the region contains a random sequence, which is similar to scrm(CTG)₇₀ that represents a random sequence. The results cannot exclude the possibility that RPA and Rad51 enrichment will be increased at the (CTG)₇₀ repeat tract. Thus, it is essential to examine the enrichment of the proteins directly on the repeat tract. This can be done with the PCR primers annealed to the 5'- and 3'-flanking regions of the repeats.
6. Figure 3, 1) The repeat instability needs to be determined using the strains with deficiency of a DNA polymerase such as pol δ or ϵ to demonstrate that the gap-filling synthesis occurred on the repeat templates resulting in repeat contraction or expansion. 2) In Figure 3f, it is not clear if there is any direct evidence in this study to show that Exo I can cleave the repeats and play a major role during

resection and processing of a repeat hairpin. 3) Lines 234-236, it was stated that "In the (CAG)₇₀ template strains, resection is impaired suggesting a CTG hairpin forms at the end of the resected strand which is resistant to processing." The sentence is confusing. It was not clear which enzyme was referred to do the resection. It was assumed that Exo I may be the one, but more evidence is needed to support its role in determining repeat instability.

7. The study revealed the mechanism underlying TNR instability induced by DSBs in yeast. How can the mechanism be extrapolated and adapted to the mammalian system, especially human cells? This point needs to be discussed and emphasized to demonstrate the impact of the study.

8. In Figure 7, the models should also include the mechanisms for the CAG repeat template to demonstrate a repeat instability in a sequence-dependent manner.

9. In "Discussion," a discussion on the contribution of this study to the research field of repeat instability and DNA repair and their potential applications in disease treatments is missing.

Minor issues

1. In "Introduction," line 39, "...in strains...", it is unclear what strain and organism were referred to here.

2. Line 383, "...enrichment of RPA 600 bp after the repeat." and Line 443, "...rather than it being an induced DSB..." are confusing. Please clarify.

Reviewer #2 (Remarks to the Author):

The authors have studied gap repair in presence of CAG/CTG repeat tract, following the induction of one DSB by HO in a yeast chromosome. The system has been derived by the integration of 70 repeats at 13 kb far from the HO cut site in the well-known strain YMV80 that has been extensively used to study DSB repair and the DNA damage checkpoint. Interestingly, they show that DNA resection and re-filling passing across CTG repeat tract lead to significantly decrease of the repeat number in the survivors. Moreover, secondary DSBs have been generated in the repeat tract, leading to larger deletions associated with loss of an essential gene. These phenotypes have been suppressed by deletion of the RAD9 gene, suggesting that the rate of CTG repeat tract exposition as single-stranded DNA might determine the formation of hairpin-like structures and their subsequent nucleolytic processing. In line with this, it is also showed that Rad51 binds and protects ssDNA at the repeats, perhaps reducing the formation of hairpins and promoting DNA resection and efficient SSA repair in this context.

Conversely, when CTG repeats are in the 5'-recessed filament, DNA resection is affected and the CAG repeats on the 3'-end template tend to small expansions.

Comments:

The topic of the work is of high interest; the experimental design is fascinating and useful to address multiple questions related to DNA repair and repeat metabolism. However, the fact that the repair system used is influenced by the kinetics of multiple events that partially overlap each other during the repair also causes confusion and difficulty to interpretate the results in more comprehensive and general view. In my opinion, the impact of the DNA resection kinetic on the formation and stability of the hairpins is a key point that requires further support.

As better indicated below, some experiments need repetition and statistical analysis; quantification of the gel band signals in the Southern blots is necessary.

Please, below specific comments and suggestions.

1) Introducing the genetic system, I think that it is important to explain why the repeat tract was inserted at 13 kb far from the break, and not closer or at more distal positions. For instance, why not at 5kb or 20kb? How does the position of the repeats/hairpins impact on the obtained results? The same is true for the number of the repeats: should we expect less effect on viability and repeat

contraction in a strain carrying a reduced number of (CTG) repeats in the template? This is also important to test in the rad9 and rad51 mutant cells (see below).

2) In Figure S1a, and in related Southern blots in Figure 2, quantification of the gel band signals is necessary. What is the extent of the SSA product reduction in the CTG template strain? Repetitions of the Southern blot analysis and related experiments are recommended to confirm and consolidate the results.

3) Results in Figure S1c and 2b. I do not see comments for the WB analysis in the CTG/CAG template strains. To me Rad53 phosphorylation appears prolonged in the CTG template strain. Is that correct? If ssDNA increases and persists longer in the CTG template cells (Figure 2b), why would be not the checkpoint higher in those cells?

4) ssDNA analysis by PCR in Figure 2 (and in other figures) should be supported by statistics. In certain cases, data variation is huge. The same comment for the ChIP analysis.

5) Regarding conclusions on resection kinetic in different strains, I'm confused because there is a tight interplay among the different repair processes (resection, sequence annealing, DNA re-synthesis), causing difficult interpretation of the results.

If possible, I think that more confident conclusion on resection kinetic can be obtained in a context in which the DSB repair is prevented by the deletion of the U2 cassette on the telomere-proximal side of the break (left side in the scheme of Figure 1). This is also important for the resection analysis in rad51 mutant cells (see below).

6) Figure 4. How is the secondary DSB induced at the hairpin structures? Is the breakage due to intrinsic fragility of the hairpins or is it mediated by specific nuclease(-s)? In case, what could be the expectation by limiting the action of the nucleolytic activity responsible for the secondary DSB formation: suppression of the cell lethality as shown in rad9 cells, or hyper-accumulation of unprocessed hairpins, also leading to adverse events and cell lethality (as perhaps shown in rad51 cells)? (See also point 9 below)

7) Figure 4. Genetic data clearly show that cell lethality is due to the loss of essential gene NFS1 in the CTG template strain. It is proposed that BIR between the HPH and NAT loci should be induced by the formation of a secondary DSB at the CTG repeat tract. However, it is not clear to me why SSA process is not taken into consideration to explain the Promoter/Terminator recombination events. Indeed, the ssDNA analysis after the repeat tract (Figure 4e) looks weak evidence of BIR.

8) Figure 4d. Quantification of the gel band signals is important here. Promoter recombination signals are visible in the scrm (CTG) control strain, however secondary break signals are not. Are there additional Southern blot analyses to show?

9) Figure 5 and the rad9 issue. Considering the multiple roles played by Rad9 in checkpoint signaling and DNA repair, it will be important to confirm and expand the proposed model testing additional mutants leading to faster/slower resection. For instance, deletion of dot1 reduces resection barrier, while deletion of fun30 increases it. Do they have any effects on repeats contraction/expansion and other relevant phenotypes? It will be also very interesting to test the contribution of nucleases/helicases involved in resection (sae2, exo1, sgs1, dna2, ...). Is it possible that they might be also involved in hairpin metabolism? Of interest, Rad9 was recently shown to limit Mph1 and Sgs1 recruitment on resected DNA: what about increased helicase activity over the CTG repeat tract as an alternative mechanism of the rad9-mediated suppression?

I have a technical concern regarding experiments in rad9 cells: to avoid possible effects due to cell cycle transition, I think that more confident results should be obtained in cells blocked in G2/M by nocodazole. This might also eliminate the contribution of DNA replication process from nearby origin. Figure 5d, quantification of the gel band signals is important here, also because the intensity of the

signals is very different with respect Figure 4d. Perhaps longer exposure is necessary here.

10) Figure 6 and the rad51 issue. In theory, deletion of Rad51 might block BIR between HPH and NAT loci, fostering the repair through SSA. However, deletion of rad51 increases cell lethality, perhaps through totally different mechanism that is explained by results in Figure 6.

As a control, I think it is important to show that NFS1 complementation assay does not rescue lethality of rad51 cells. By the way, what is the level of secondary DSB at the CTG repeats in rad51 cells? Quantification of the Southern blot in figure S6a is important, together with a longer exposure of the blots. Can we exclude a role of secondary DSB formation at the repeat tract in the lethality of rad51 cells? For instance, is it possible that rad51 cells accumulate irreparable DSBs at the repeats, leading to hyperactivation of Rad53 and cell lethality? On this turn, how is the double mutant rad9 rad51 in term of vitality and repeat contraction/expansion? Is it possible to show the terminal phenotype of rad51 cells, following HO break? Do they remain blocked in G2/M (similar to adaptation-defective cells) or start dividing forming microcolonies of few cells?

Reviewer #3 (Remarks to the Author):

The manuscript provides new mechanistic insights into triplet repeat instability during repair of large DNA gaps in yeast. Some of the findings, such as contraction bias of CTG template and expansion bias of CAG templates, had been inferred in earlier studies, but the manuscript does break new ground by testing specific models of instability and looking at DNA intermediates during processing. For example, the data in Fig3b are especially interesting because the expansion bias seen under DSB conditions mimics to some extent what is seen in long triplet repeat tracts in humans and mice. The second cut site in single stranded CTG tracts is also novel and interesting. The interpretation of these data provides some insights into mechanistic behavior of the DNA strands and their interaction with repair proteins that might apply, in general, to mammals as well as yeast.

The one major issue with the manuscript is the control experiment looking for hyperactivation of Rad53. Lines 123-125 claim there was no persistent hyperactivation of Rad53, but Fig S1c is hard to interpret due to frowning on the gel. Also the figure does not provide a positive control for phosphoRad53. Better data is required to confirm this key control.

Minor points

1. Lines 133-134 claim that resection, annealing and fill-in synthesis are likely occurring "concurrently". This is an overstatement since, from the data presented, as much as two hours may be elapsing between resection and fill-in.
2. The Rad51 enrichment data in SFig2b are a bit confusing in light of the earlier claims of the paper that the assay is SSA (abstract) but that SSA is Rad51-independent (lines 139-140). A bit more explanation is needed to make this point more clear at this point of the manuscript. The SSA-BIR description later is helpful, but some clarity in the text about SFig2b is needed.
3. Line 253 mentions dark green boxes in Fig 4a, but I think this should read purple boxes.
4. Lines 431-4 describe experiments from HD mice (ref 6). A similar idea was proposed for myotonic dystrophy based on mathematical modelling (10.1093/hmg/dd5059). It would be useful to consider citing the DM paper in addition.

REVIEWER COMMENTS

Reviewer #1 (Remarks to the Author):

Manuscript # NCOMMS-22-11274

Trinucleotide repeat (TNR) instability is associated with human neurodegenerative diseases and cancer and can occur during DNA replication and repair. HR is one of the major mechanisms that underlie large repeat instability. However, the precise mechanism by which HR mediates repeat instability remains unclear and needs to be elucidated. One of the critical steps in HR is strand resection, which can result in a large single-strand DNA gap. The process may allow the generation of secondary structures in the repeat tracts leading to repeat instability during double-strand break repair. In this study, the authors developed a system to detect the resection and ssDNA gap-filling on CAG or CTG repeats. Using the system, they found that when CTG repeats served as the single-stranded template, the repair of the gap promoted repeat deletions. In contrast, the repair resulted in repeat expansion when CTG repeats on the resected strand. They further demonstrated that Rad9 prevented resection and slowed down the gap-filling synthesis, and Rad51 inhibited repeat deletion. The results suggest that different repeat DNA sequences in the single-strand DNA gap govern the repair rate and repeat instability. The study provided new evidence to support a mechanism underlying trinucleotide repeat instability during HR through gap-filling synthesis. The conclusions of the study are supported by the comprehensive experimental data. However, some major and minor issues shown below need to be addressed.

Major issues

1. In “Abstract,” it is unclear why it is important to make the discoveries reported in this study and how Rad9 and Rad51 coordinated with DNA sequences to determine repeat deletion or expansion. In fact, in “Introduction,” the authors gave an excellent introduction on the research gap and provided the justifications for their study. The same points need to be also presented in the “Abstract.”

We have added the research gap and study justification to the abstract:

Homologous recombination (HR) is one cause of repeat instability and we hypothesized that gap filling was a driver of repeat instability during HR. To test this...

2. In Fig. 2c, it is surprising that the scrm (CTG)₇₀ template exhibited a higher percentage of ssDNA than (CAG)₇₀. Supplementary Fig. 2a showed that the scrm(CTG)₇₀ template exhibited the highest percentage of ssDNA. Why did this happen? Does this mean the gap-filling synthesis on the region upstream of the repeats was facilitated by the repeats because it had less ssDNA?

In Fig. 2c, the (CAG)₇₀ orientation indeed has a significant decrease in the amount of ssDNA *after the repeat* compared to the scrm(CTG)₇₀ template, which is statistically significant 8 hours post-DSB induction (stats now added). We believe this is biologically significant and conclude that when (CAG)₇₀ is the template, CTG hairpins on the resected strand act as a barrier to resection (right side of Fig. 2d). This is supported by 2 pieces of data: (1) the parental strain (Supplementary Fig. 1c) shows the same profile as the scrm(CTG)₇₀ template control (indicating it isn't increased resection in the scrm control but decreased in the (CAG)₇₀ template strain), and (2) there is no significant difference between % ssDNA *before the repeat* in the scrm(CTG)₇₀ and (CTG)₇₀ template strains at any timepoint (Supplementary Fig. 2a; stats now done). As noted, Supplementary Fig. 2a *does* show faster fill-in for the (CAG)₇₀ template strain, which is statistically significant at later time points (stats now added).

Yes, we agree this could mean that gap-filling synthesis on the region upstream of the repeats was indirectly facilitated by the repeats – based on our data we think it is because there was a smaller gap due to the resection barrier. We now outline this conclusion more clearly in the text (see below).

Also, in lines 165-166, the conclusion is that a CTG hairpin on the 5'-recessed strand prevents resection. Was the conclusion drawn based on the results of Supplementary Fig. 2a? There is no clear discussion about this. A justification and rationale need to be provided here.

We have now improved our justification and rationale for this conclusion as follows. See also lines 163-169 for an improved initial description of this data.

In strains that have a (CAG)₇₀ fill-in template, we observed reduced ssDNA after the repeat tract (Fig. 2c) and faster gap filling whether measured before (Supplementary Fig. 2a) or after (Fig. 2c) the repeat. We propose that in this case, the CTG hairpin on the 5' recessed strand acts as a barrier to resection, which decreases the size of the ssDNA gap due to the hairpin-impaired resection (Fig. 2d, right panel).

3. To provide direct evidence to support the roles of gap-filling synthesis by DNA polymerases *in vivo*, it is critical to test the effects of gap-filling synthesis on the accumulation of ssDNA using a DNA polymerase deficient strain.

We agree this is an important goal and it is the subject of a follow-up paper in preparation. Testing all the possible DNA polymerases using various non-lethal mutations or depletions is beyond the scope of this study.

4. It is also critical to show ssDNA accumulation on the (CAG)₇₀, (CTG)₇₀, and scrm(CTG)₇₀ directly rather than just show them on the upstream and downstream strands of the repeat tracts. This can be done using the PCR primers annealed at the random DNA sequences at both the 5'- and 3'-flanking regions of the repeats. The P1 and P2 in Fig. 1 may be used as the primers for the experiments.

Due to the tricky nature of CAG/CTG repeat amplification and the difficulty in manipulating qPCR master mixes we have been unable to amplify across the repeat tract in a quantitative manner. Thus, we are unable to quantify ssDNA directly at the repeat. We did use a qPCR primer set closer to the repeat for ChIP amplification (below) and the amount of RPA enrichment detected using primers 60 bp from the repeat is not different than the amount of enrichment detected using primers 600 bp from the repeat. We conclude that the primer sets before and after the repeat tract used in the resection assay are able to accurately measure impairments due to the repeat.

5. In Supplementary Fig. 2, it is not surprising that no significant difference was detected in the enrichment of RPA and Rad51 on the 600 bp fragment that is located at downstream of the repeats. Because the region contains a random sequence, which is similar to scrm(CTG)₇₀ that represents a random sequence. The results cannot exclude the possibility that RPA and Rad51 enrichment will be increased at the (CTG)₇₀ repeat tract. Thus, it is essential to examine the enrichment of the proteins directly on the repeat tract. This can be done with the PCR primers annealed to the 5'- and 3'-flanking regions of the repeats.

As described above, though detecting qualitative repeat tract size differences is possible, we are unable to amplify across the repeat tract in a quantitative manner. However, we repeated the ChIP qPCR using a primer set within 60 bp of the repeat tract and found similar enrichment trends as found 600 bp from the repeat. We used a t-test to make comparisons between enrichment at the 60 bp and 600 bp locations and saw no significant differences, suggesting both locations have comparable amounts of RPA and Rad51 within the detection limits of this assay (p values are listed in table 6). For all ChIP data sets, we now show both locations tested.

6. Fig. 3, 1) The repeat instability needs to be determined using the strains with deficiency of a DNA polymerase such as pol δ or ϵ to demonstrate that the gap-filling synthesis occurred on the repeat templates resulting in repeat contraction or expansion.

We agree this is an important goal and is the subject of a follow-up paper. So far we have confirmed that gap-filling synthesis by polymerases is a cause of repeat expansions and contractions, and there are interesting aspects that require a more complete analysis than can be covered in this paper.

2) In Fig. 3f, it is not clear if there is any direct evidence in this study to show that Exo I can cleave the repeats and play a major role during resection and processing of a repeat hairpin.

In our assay, resection is initiated from a DSB and our assumption is that the canonical resection machinery continues through the repeat. As the repeat is situated ~13 kb away from the DSB, we expect long range resection proteins like Exo I to be needed. See pt. 3 below for additional rationale for not testing proteins required for long range resection.

3) Lines 234-236, it was stated that “In the (CAG)₇₀ template strains, resection is impaired suggesting a CTG hairpin forms at the end of the resected strand which is resistant to processing.” The sentence is confusing. It was not clear which enzyme was referred to do the resection. It was assumed that Exo I may be the one, but more evidence is needed to support its role in determining repeat instability.

Experimentally, it is difficult to address which nuclease is impaired by the repeat as deletion of long range resection proteins will impact resection kinetics that are initiated from the DSB. It’s been previously demonstrated that deletion of resection proteins such as Sgs1/Dna2 and Exo1 slow resection and that there is compensation between the mid and long range resection epistasis groups¹⁻³. In addition, it’s been shown that loss of resection proteins leads to decreased viability in our parent strain due to impaired repair⁴. As our goal is to test fill-in, we need resection and repair to occur efficiently enough that the surviving colonies can be assayed for phenotypes related to gap filling.

We have edited the text for clarification and it now reads:

In the (CAG)₇₀ template strains, our data indicate a difficulty resecting through the repeat tract, suggesting a barrier such as a CTG hairpin on the 5’ end of the resected strand. This hairpin could be resistant to processing by the canonical long-range DSB resection machinery (consisting of Exo1 and Dna2/Sgs1 in yeast¹⁻³) and/or other endo-exonucleases.

7. The study revealed the mechanism underlying TNR instability induced by DSBs in yeast. How can the mechanism be extrapolated and adapted to the mammalian system, especially human cells? This point needs to be discussed and emphasized to demonstrate the impact of the study.

We have added additional commentary in the discussion on how our data may relate to mammalian systems:

Our results indicate that gap repair is mutagenic in the context of a structure-forming DNA sequence. This mechanism could be particularly relevant to cancer cells which have lots of gapped DNA⁵. It should also be considered when designing target sites for gene editing: if a repetitive, structure forming sequence is near a DSB target locus, it is possible that a secondary break and alternative repair could result. Our findings indicate that the success and accuracy of repair is influenced by the sequence context where repair is occurring and illustrate the danger of exposing ssDNA within repetitive sequences during gap repair.

8. In Fig. 7, the models should also include the mechanisms for the CAG repeat template to demonstrate a repeat instability in a sequence-dependent manner.

We chose to put the model for repeat instability in a sequence dependent manner in Fig. 3 e,f because we think it adds clarity to the story at a necessary point. After Fig. 3, we delve into a different arm of the study: understanding the loss of viability and

chromosomal fragility when CTG is on the template strand. The model in Fig. 7 reflects that latter part of the story presented in Fig. 4 (the two break hypothesis), Fig. 5 (rescue by loss of Rad9) , and Fig. 6 (requirement for Rad51).

Discussions of the instability model (Fig. 3 e, f) can be found on the results section of Fig. 3 (ln 218-241; p. 10-11) and in the discussion (p. 19-20), where we refer back to the models in Fig 3 e, f.

9. In “Discussion,” a discussion on the contribution of this study to the research field of repeat instability and DNA repair and their potential applications in disease treatments is missing.

We had added additional impact statements relating our data to TNR diseases, cancers and gene editing to the discussion:

TNR diseases:

These small-scale expansions could be like those that occur during gap repair in non-dividing human cells, such as neurons, that undergo stepwise somatic expansions which advance disease onset in TNR expansion disorders such as Huntington’s disease⁶. Modeling of the expansion bias observed in humans using a large sample of blood DNA samples collected from Myotonic Dystrophy type 1 (DM1) patients predicted that hundreds of small expansion and contraction events accumulate over the lifespan of a hematopoietic stem cell, and the rate was consistent with occurrence due to DNA damage and repair rather than replication⁷. Thus, our data provide a mechanism for small TNR expansion events that can result in worsening of disease phenotypes in somatic cells.

Cancer cells and gene editing:

Our results indicate that gap repair is mutagenic in the context of a structure-forming DNA sequence. This mechanism could be particularly relevant to cancer cells which have lots of gapped DNA⁵. It should also be considered when designing target sites for gene editing: if a repetitive, structure forming sequence is near a DSB target locus, it is possible that a secondary break and alternative repair could result. Our findings indicate that the success and accuracy of repair is influenced by the sequence context where repair is occurring and illustrate the danger of exposing ssDNA within repetitive sequences during gap repair.

Minor issues

1. In “Introduction,” line 39, “...in strains...”, it is unclear what strain and organism were referred to here.

Sentence has been clarified to: Additionally, in yeast containing a long CAG repeat tract on a yeast artificial chromosome (YAC), recombination-dependent expansions and contractions were observed in strains carrying mutations in proteins important in DNA repair or replication.

2. Line 383, "...enrichment of RPA 600 bp after the repeat." and Line 443, "...rather than it being an induced DSB..." are confusing. Please clarify.

Ln 383 (now line 428): Sentence has been clarified to: Confirming the delayed resection phenotype in the *rad51*Δ mutant, delayed enrichment of RPA was also observed (Fig. 6f, Supplementary Fig. 6e)

Ln 443 (now Ln 518): Sentence has been clarified to: This second model for repeat contractions is similar to one proposed for Cas9-induced breaks at an expanded CAG/CTG repeat which repair via end joining or SSA⁸. However, there are differences, as the Cas9-induced break is directly targeting the double-stranded repeat whereas in our assay the repeat tract becomes single-stranded due to resection first and then is converted to a break.

Reviewer #2 (Remarks to the Author):

The authors have studied gap repair in presence of CAG/CTG repeat tract, following the induction of one DSB by HO in a yeast chromosome. The system has been derived by the integration of 70 repeats at 13 kb far from the HO cut site in the well-known strain YMV80 that has been extensively used to study DSB repair and the DNA damage checkpoint. Interestingly, they show that DNA resection and re-filling passing across CTG repeat tract lead to significantly decrease of the repeat number in the survivors. Moreover, secondary DSBs have been generated in the repeat tract, leading to larger deletions associated with loss of an essential gene. These phenotypes have been suppressed by deletion of the RAD9 gene, suggesting that the rate of CTG repeat tract exposition as single-stranded DNA might determine the formation of hairpin-like structures and their subsequent nucleolytic processing. In line with this, it is also showed that Rad51 binds and protects ssDNA at the repeats, perhaps reducing the formation of hairpins and promoting DNA resection and efficient SSA repair in this context.

Conversely, when CTG repeats are in the 5'-recessed filament, DNA resection is affected and the CAG repeats on the 3'-end template tend to small expansions.

Comments:

The topic of the work is of high interest; the experimental design is fascinating and useful to address multiple questions related to DNA repair and repeat metabolism.

However, the fact that the repair system used is influenced by the kinetics of multiple events that partially overlap each other during the repair also causes confusion and difficulty to interpretate the results in more comprehensive and general view. In my opinion, the impact of the DNA resection kinetic on the formation and stability of the hairpins is a key point that requires further support.

As better indicated below, some experiments need repetition and statistical analysis; quantification of the gel band signals in the Southern blots is necessary.

Please, below specific comments and suggestions.

1) Introducing the genetic system, I think that it is important to explain why the repeat tract was inserted at 13 kb far from the break, and not closer or at more distal positions. For instance, why not at 5kb or 20kb?

The decision of where to insert the repeat tract was made by a variety of different factors relating to the previously determined kinetics of the break induction and repair process and finding a genomic region of Chromosome III where we could insert the repeat without affecting expression of essential genes.

Edited to include: To study instability of the CAG/CTG repeat during fill-in synthesis, TNR tracts were integrated into the *ILV6* locus, a non-essential gene ~13 kb away from the DSB on the centromere-proximal side. Because resection occurs equally on both sides of the break¹, this distance ensured that the repeat tract would be fully single-stranded by the time the U2 region on the other side of the HO cut site was single stranded. This distance facilitated kinetic analysis as it provided time to follow both the resection and repair steps.

How does the position of the repeats/hairpins impact on the obtained results?

This is an active future direction of the work and is beyond the scope of this paper.

The same is true for the number of the repeats: should we expect less effect on viability and repeat contraction in a strain carrying a reduced number of (CTG) repeats in the template?

Past work in the field has shown a length dependent relationship to rates of fragility and instability⁹. To address this question in the context of gap repair we constructed a (CTG)₃₀ strain, which is a length close to, but below the instability threshold, and tested viability and instability. As expected, there was a 3.5 fold increase in viability in the (CTG)₃₀ strain compared to the (CTG)₇₀ strain (Fig. 1c). We also measured instability in the (CTG)₃₀ strain and found no significant increase in expansions or contractions in the DSB condition compared to the no break condition (now added to Supplementary Fig. 3c, d). Together this supports the idea that fragility and instability during gap filling are linked to repeat length.

Text added ln115-118: Previous work has shown that TNR repeat fragility is length dependent, with longer repeats having higher rates of breakage⁹. To confirm that the loss in viability is due to repeat length, we constructed a (CTG)₃₀ assay strain and found no significant defect in viability compared to the scrambled control strain (Fig. 1c).

Text added ln232-239: To confirm that gap fill-in mediated instability was a function of the structure-forming ability of the repeat tract, we determined the instability of both the (CTG)₃₀ and the scrambled control strains (Supplementary Fig. 3c-f). Though some instability exists for (CTG)₃₀ and the scrambled control, expansion and contraction frequencies were not significantly different between the no-break and break conditions: (CTG)₃₀ (Supplementary Fig. 3d, f). Taken together, this suggests repeat length and ability to form DNA hairpins are prime factors that drive repeat instability during gap-filling.

This is also important to test in the *rad9* and *rad51* mutant cells (see below).

As we describe above, wildtype (CTG)₃₀ strains have increased viability and no gap fill-in related instability. We think it is unlikely anything additional would be gained from testing a *rad9*Δ or *rad51*Δ mutant in the (CTG)₃₀ strain.

2) In Supplementary Fig. 1a, and in related Southern blots in Fig. 2, quantification of the gel band signals is necessary. What is the extent of the SSA product reduction in the CTG template strain? Repetitions of the Southern blot analysis and related experiments are recommended to confirm and consolidate the results.

We have now quantified the bands on the Southern blots for at least 2 repetitions of each time course for each strain and done statistical analysis. Description of Southern quantification can be found in Methods (ln 632-640). Number of replicates is listed in figure legends and all raw percentages and p-values can be found in supplemental table 7. All timepoints had significantly lower repair product accumulation in the (CTG)₇₀ template strain compared to the scrm(CTG)₇₀. The (CAG)₇₀ template strain had no significant differences compared to the scrm(CTG)₇₀ control. This data is now presented in Fig. 2b. Comparison of repair product formation between the scrm(CTG)₇₀ and parental strain are shown in Supplementary Fig. 1b.

3) Results in Supplementary Fig. 1d and 2c. I do not see comments for the WB analysis in the CTG/CAG template strains. To me Rad53 phosphorylation appears prolonged in the CTG template strain. Is that correct? If ssDNA increases and persists longer in the CTG template cells (Fig. 2b), why would be not the checkpoint higher in those cells?

The westerns to detect Rad53 phosphorylation have been repeated with a positive control (as requested by reviewer 3) and replace the old data in Fig S1d. The new higher quality data show more clearly that there is no difference in activation or extinction of the DNA damage checkpoint in the (CTG)₇₀ strain relative to the parental, scrm(CTG)₇₀ and (CAG)₇₀ strains.

As for the explanation as to how it relates to Fig. 2c: We do not see a difference in checkpoint activation or extinguishment in the (CTG)₇₀ template strains. We believe that the increased ssDNA in later timepoints post-DSB induction in this strain is likely due to the second break that occurs in the CTG strand (depicted in Fig. 4a). Our data show that this leads to repair using alternative exposed promoter homologies by a Rad51-dependent process, likely BIR (see Supplementary Fig. 4e, Supplementary Fig. 6c). Therefore, a reasonable explanation is that there is delayed second strand fill-in, which is a well-established step of BIR¹⁰, which is discussed in lns 331-339. The kinetics of SSA and BIR are similar in this strain¹¹, which could explain why the switch to BIR did not result in delayed deactivation of the DNA damage checkpoint.

4) ssDNA analysis by PCR in Fig. 2 (and in other figures) should be supported by statistics. In certain cases, data variation is huge. The same comment for the ChIP analysis.

Previous studies from the Haber, Symington and Ira labs have not used a statistical test to determine differences in resection and usually plot the mean and standard deviation as we also show. Also data variation is inherent to this type of assay as it consists of at least 2 repeated independent time courses. However, in response to this request we utilized a Student's t-test which is an appropriate way to measure differences between individual time-points as it is a comparison of means. Statistical significance stars have been added to all graphs plotting % ssDNA and a comprehensive listing of all p values have been added to supplemental table 3.

Similarly, we have used a Student's t-test to compare paired timepoints for quantified repair products from the Southern blot time courses of each relevant strain and added appropriate star designations.

We unfortunately had different pull-down efficiencies in our ChIP experiment repetitions, but the pattern of timing of ssDNA binding protein recruitment was consistent between trials. To help interpret trends, our graphs now use differential symbols to denote experiment 1 (●) and experiment 2 (◆). As requested by reviewer 1, we have added an additional closer amplicon for our ChIP analysis and used the Student's t-test to compare enrichment of RPA and Rad51 between primer sets (in table 6). This adds an additional level of confidence in the data and shows that the *timing of recruitment* of [RPA or Rad51] is consistent between trials, which is what we are analyzing (we are not making conclusions about overall levels of RPA or Rad51 between strains, for example).

These statistical tests aren't the whole picture, as they only indicate significance of individual data points that are part of an entire time course and it's important to consider the trend of the curve more broadly in addition to the distinct time point data.

5) Regarding conclusions on resection kinetic in different strains, I'm confused because there is a tight interplay among the different repair processes (resection, sequence

annealing, DNA re-synthesis), causing difficult interpretation of the results.

If possible, I think that more confident conclusion on resection kinetic can be obtained in a context in which the DSB repair is prevented by the deletion of the U2 cassette on the telomere-proximal side of the break (left side in the scheme of Fig. 1). This is also important for the resection analysis in *rad51* mutant cells (see below).

We agree that this would be a good experiment if our main goal was to study resection kinetics. But our main goal was to study the gap repair process, and how it occurs over a structure-forming repeat (repeat instability and healing kinetics). Loss of the U2 homology means neither event can be measured. Also, it has already been published that resection is faster in *rad9Δ* mutants¹², and our data recapitulates that result. Our result that resection is slower in the absence of Rad51 is novel, but it is not a main point of our paper. Given that what we would learn from this experiment is minimal and not directly related to our goals, and that it would necessitate re-making all our strains and collecting time course data, we do not think it would be impactful enough to justify the time and effort.

6) Fig. 4. How is the secondary DSB induced at the hairpin structures? Is the breakage due to intrinsic fragility of the hairpins or is it mediated by specific nuclease(-s)? In case, what could be the expectation by limiting the action of the nucleolytic activity responsible for the secondary DSB formation: suppression of the cell lethality as shown in *rad9* cells, or hyper-accumulation of unprocessed hairpins, also leading to adverse events and cell lethality (as perhaps shown in *rad51* cells)? (See also point 9 below)

This is an excellent question. There are data to suggest that DNA hairpins and other secondary structures are a target of endonucleases. In addition, in our model we hypothesize that delayed recruitment of RPA result in hairpin formation and later nuclease cleavage. The Symington lab showed that depletion of RPA resulted in impaired resection and repair due to hairpin capped formation which are processed by Mre11/Sae2¹³. As such, we have tested a variety of nucleases that are known to target DNA structures with the assumption that if they were the nuclease, they would reduce the loss in viability in the (CTG)₇₀ strain. Creation of *MLH1*, *SLX1* and *MUS81* single mutants did not result in increased viability. We have now added these data to Supplementary Fig. 4e and added a discussion of this issue to the text:

Ln324-328: We deleted structure specific endonucleases that have been previously shown to target DNA hairpins, such as Mlh1, Mus81 and Slx1^{14,15,16} and found no increase in viability compared to wildtype (Supplementary Fig. 4d), suggesting either there is redundancy between nucleases or a that different nuclease is responsible for the breakage at the CTG repeat.

Another possible candidate is Mre11, which is unfortunately challenging to test in the context of this gap repair assay, as it is required for the initiation of resection and deletion of components of the MRX complex (*rad50Δ*, *sae2Δ*) leads to low viability (~5-10%^{4,17}) in the parent strain, YMV80, which requires extensive resection to heal the induced DSB. The nuclease dead version of Mre11 (*mre11-D56N* allele), is also complicated by low viability (~30%) in YMV80⁴. The expected *mre11-D56N* viability is

only 10% higher than our wildtype (CTG)₇₀ strain (~20%) so it would be difficult to obtain a clear result of genetic rescue. Despite this we attempted to make the *mre11-D56N* mutant, but interestingly, attempts to make this mutant in the (CTG)₇₀ strain failed, suggesting that the strain may be inviable when there is an integrated repeat. We screened over 100 colonies using a pop-in/pop-out strategy and an additional 50 in a strain with a *MRE11* covering plasmid and never obtained a mutant with an intact repeat tract. Further confirming that this may be a repeat specific phenotype, 4 mutants were obtained after screening only 6 colonies in the *scrm*(CTG)₇₀ control, which has viability similar to what has been previously published⁴. Based on published results, Mre11 is predicted to be required to process hairpin-capped ends that form when a break occurs in the CTG repeat tract. We think that without this processing, repair of the CTG break is inhibited which would be lethal when occurring at the ChrIII location used in this study.

A discussion of the possibility of Mre11 being the nuclease has been added to the discussion (In 529-535): One nuclease that could be targeting the CTG repeat is Mre11 as it is required for processing hairpin-capped ends¹⁸ and creates breaks at inverted repeats that form hairpins on ssDNA tracts that occur during lagging strand replication¹⁶. However, Mre11 is required for the initiation of resection, and loss of the MRX complex or Mre11's nuclease activity result in significantly decreased viability in this strain^{4,17}, preventing a test to determine whether Mre11 is the nuclease responsible for the break at the CTG repeat tract.

7) Fig. 4. Genetic data clearly show that cell lethality is due to the loss of essential gene *NFS1* in the CTG template strain. It is proposed that BIR between the HPH and NAT loci should be induced by the formation of a secondary DSB at the CTG repeat tract. However, it is not clear to me why SSA process is not taken into consideration to explain the Promoter/Terminator recombination events. Indeed, the ssDNA analysis after the repeat tract (Fig. 4e) looks weak evidence of BIR.

While it is true that repair between the promoter/terminator can occur via both SSA and BIR, our data support BIR as a larger contributor due to the persistence of ssDNA at later timepoints which is exacerbated in a *rad51Δ* strain (Fig. 6e) and loss of this repair product in the *rad51Δ* strain (Fig. 6d). To further support the BIR hypothesis, we did *NFS1* complementation experiments in the *rad51Δ* strains with the hypothesis that if the promoter/terminator repair was BIR, then we would not see the small colony rescue as we see in wildtype cells. Supportive of this hypothesis, we see minimal recovery of small colonies in the *rad51Δ* mutant (Supplementary Fig. 6f,g). Thus, while both BIR and SSA can rescue secondary DSBs at the CTG repeat, we conclude that BIR is more frequent.

This is now described in text: In413-423: To better determine whether the addition of a repeat tract changes repair kinetics in the *rad51Δ* mutant, we followed the time course of the U2 repair reaction by Southern blot (Fig. 6c). Consistent with previous work¹¹, the U2 repair product was significantly delayed in the *rad51Δ* mutant strains (Fig. 6c), first

appearing around 8 hours and then increasing slowly from 10-24 hours (Supplementary Fig. 6a). Breaks that occur at the CTG repeat tract appear when the DNA first becomes single-stranded at hour 6 and persist through hour 12 in the *rad51Δ* mutant (Supplementary Fig. 6b, c). Alternative recombination between the TEF promoters in the *rad51Δ* mutant is also significantly reduced compared to wildtype in the (CTG)₇₀ template strain (Fig. 6d, Supplementary Fig. 6b) suggesting that Rad51-dependent repair is driving promoter recombination when there is a break at the CTG repeat.

8) Fig. 4d. Quantification of the gel band signals is important here. Promoter recombination signals are visible in the *scrm* (CTG) control strain, however secondary break signals are not. Are there additional Southern blot analyses to show?

We have now quantified the frequency of repair product using promoter homology and DSB formation at the CTG tract in the *scrm*(CTG)₇₀ (n=2) and (CTG)₇₀ (n=4) strains for multiple blots and included the data in Supplementary Fig. 4d (Percent CTG DSB formation) and Supplementary Fig. 4c (promoter homology repair product). Supportive of our previous conclusions, we see a significant increase in CTG DSB formation in the (CTG)₇₀ template strain at hour 4 and 6 post break induction and a concomitant increase in promoter homology repair product from hours 6-24. For all blot measurements there is an n of at least 2, exact number of replicates for each measurement are listed in the figure legends

This is discussed in In308-312: We observed a unique band in the (CTG)₇₀ template strain that corresponds to the size expected if the break occurred within the (CTG)₇₀ repeat tract (Fig. 4d). Measurement of the signal of the break at the (CTG)₇₀ repeat showed a significant increase compared to the *scrm*(CTG)₇₀ at hour 4 and 6 post-DSB induction (Fig. 4d).

9) Fig. 5 and the *rad9* issue. Considering the multiple roles played by Rad9 in checkpoint signaling and DNA repair, it will be important to confirm and expand the proposed model testing additional mutants leading to faster/slower resection. For instance, deletion of *dot1* reduces resection barrier, while deletion of *fun30* increases it. Do they have any effects on repeats contraction/expansion and other relevant phenotypes?

As our assay relies on extensive resection in order for repair to be initiated, we cannot test the role of Fun30 in our assay. It's been previously noted that Fun30 is required for resection and a *fun30Δ* mutant has a resection defect¹⁹. We suspect that a *DOT1* deletion might phenocopy *rad9Δ* but resection kinetics and chromatin remodeling is beyond the scope of this study. This would be an interesting future direction given our *rad9Δ* findings.

It will be also very interesting to test the contribution of nucleases/helicases involved in resection (*sae2*, *exo1*, *sgs1*, *dna2*, ...). Is it possible that they might be also involved in hairpin metabolism?

It has been shown that several nucleases and helicases are responsible for processing hairpins. As our assay relies on extensive resection for both repair and later gap filling through the repeat tract, we cannot test many of these proteins as they are also required for resection. For example, Mre11/Sae2, Exo1, Sgs1, and Dna2 are all required for resection initiation or long range resection^{1,17}. For nucleases that have no known role in resection we have included them in Supplementary Fig. 4a and they are discussed in the revised text (Ln324-328): We deleted structure specific endonucleases that have been previously shown to target DNA hairpins, such as Mlh1, Mus81 and Slx1^{14,15,16} and found no increase in viability compared to wildtype (Supplementary Fig. 4d), suggesting either there is redundancy between nucleases or a that different nuclease is responsible for the breakage at the CTG repeat.

Of interest, Rad9 was recently shown to limit Mph1 and Sgs1 recruitment on resected DNA: what about increased helicase activity over the CTG repeat tract as an alternative mechanism of the rad9-mediated suppression?

This is an interesting idea. If Rad9 is limiting recruitment of these helicases, there might be more CTG hairpin unwinding by Mph1 or Sgs1 in *rad9Δ* cells, which fits with our data of less repeat contractions. We have added this possibility to our discussion.

Ln 556-563: Rad9 may also influence loading of other repair factors that play a role in repeat stability. For example, Rad9 limits recruitment of the helicases Mph1 and Sgs1 to resected DNA²⁰. Since non-B form DNA structures that form due to TNRs are unwound by helicases to prevent fragility and instability²¹, another possibility is that the Rad9 mediated rescue is due to improved unwinding of the CTG hairpin.

I have a technical concern regarding experiments in *rad9* cells: to avoid possible effects due to cell cycle transition, I think that more confident results should be obtained in cells blocked in G2/M by nocodazole. This might also eliminate the contribution of DNA replication process from nearby origin.

Because of the checkpoint, the repair events we are monitoring are predominantly occurring in G2/M and no replication is occurring. Replication-associated events are our “no break control” background. We compare all our data to the no-break control to focus on the gap repair-specific events. Therefore, if there is some checkpoint escape in *rad9Δ* mutants leading to a replication effect, this will be accounted for by the no break control. Indeed some suppression of contractions is observed in the no break condition, though significantly less than in the DSB break condition (Fig 5b). This suggests that the main suppressive effect is due to the accelerated resection and gap repair occurring in *rad9Δ* mutants.

Nonetheless, we attempted to address this question experimentally by treating wildtype and *rad9Δ* mutant cells with 15 μg/ml nocodazole for the length of time that it took non-treated cells to extinguish the checkpoint and exit G2/M arrest (~12 hours) and plated for viability. We found wildtype cells had significant viability defects and a substantial number of cells were triple budded. As the viability is negatively impacted in wildtype

cells, we are unable to test the *rad9Δ* mutant in a nocodazole block and draw significant conclusions.

Supplementary Fig. 5c, quantification of the gel band signals is important here, also because the intensity of the signals is very different with respect Fig. 4d. Perhaps longer exposure is necessary here.

It is impossible to quantify the CTG repeat break band in the *rad9Δ* mutant as it is not present, even upon longest exposures. However, we have now quantified expected repair in the *rad9Δ* mutant and percent promoter homology repair events. In the *rad9Δ* mutant in the (CTG)₇₀ template strain background we found increased U2 repair that mirrors the maximum amount of repair in the *scrm*(CTG)₇₀ strain (Fig. 5d). In addition the amount of promoter repair product in *rad9Δ* mutant in the (CTG)₇₀ template strain background decreases compared to the (CTG)₇₀ strain (Supplementary Fig. 5d). Taken together this suggests that breaks are no longer occurring at the repeat in the *rad9Δ* mutant in the (CTG)₇₀ template strain background.

New text added: Ln369-373: Consistent with faster resection and filling in, there was a significantly earlier appearance of the U2 repair product in the *rad9Δ* mutant compared to wildtype (Fig. 5d, Supplementary Fig. 5b). Further the amount of U2 repair in the *rad9Δ* mutant was significantly increased compared to the wildtype (CTG)₇₀ template strain (Fig. 5d).

Ln 377-380: Consistent with the lack of breakage at the CTG repeat tract in the *rad9Δ* mutant, there is decreased levels of TEF promoter recombination compared to wildtype for the (CTG)₇₀ template strain (Supplementary Fig. 5d).

10) Fig. 6 and the *rad51* issue. In theory, deletion of Rad51 might block BIR between HPH and NAT loci, fostering the repair through SSA. However, deletion of *rad51* increases cell lethality, perhaps through totally different mechanism that is explained by results in Fig. 6.

We do believe that deletion of Rad51 blocks BIR between the HPH and NAT loci, and the newly added quantification of the repair products support this conclusion, now presented in Figs. 6c & 6d and text lines 415-423.

The data shows that there is no increase in the amount of repair using promoter homology in the *rad51Δ* mutant in either the *scrm*(CTG)₇₀ or (CTG)₇₀ strains (Fig. 6d), suggesting this repair relies primarily on Rad51. In the *rad51Δ* mutant the expected U2 repair product is significantly delayed and this repair defect is worse in the strain with the CTG repeat and only reaches about 50% of wildtype levels (Fig. 6b). Meanwhile, the CTG breakage product persists (Supplementary Fig. 6c). Therefore, in the *rad51Δ* mutant SSA cannot fully compensate to repair the CTG break. We conclude that the loss of the BIR pathway is a main reason for *rad51Δ* mutant lethality, and that this pathway is especially important to repair the break at the CTG repeat.

By the way, what is the level of secondary DSB at the CTG repeats in *rad51* cells? Quantification of the Southern blot in Supplementary Fig. 6a is important, together with a longer exposure of the blots. Can we exclude a role of secondary DSB formation at the repeat tract in the lethality of *rad51* cells?

We have increased the exposure of the *rad51* Δ mutant blot (now in Supplementary Fig. 6b) and quantified the CTG DSB band (now in Supplementary Fig. 6c). Quantification shows a similar level of breakage in the *rad51* Δ (CTG)₇₀ strains compared to wildtype, but a longer persistence of the broken product. This suggests that it's not increased breakage that causes the decreased viability in the *rad51* Δ mutant, but rather poor repair. So yes, indirectly the secondary DSB at the repeat tract is the cause of the lethality in *rad51* Δ cells.

For instance, is it possible that *rad51* cells accumulate irreparable DSBs at the repeats, leading to hyperactivation of Rad53 and cell lethality?

We measured Rad53 hyperactivation in *rad51* Δ mutant cells in the scrambled and CTG template strains and found prolonged activation of the DNA damage checkpoint compared to wildtype (Supplementary Fig. 6i). In *rad51* Δ mutant cells, the checkpoint is activated through hour 12, but is extinguished by hour 24, while in wildtype cells, the checkpoint is mostly extinguished by hour 10 (Supplementary Fig. 1d). This mirrors the persistence of ssDNA in *rad51* Δ cells through hour 12 but reduction by 24h (Fig. 6d). Thus, while resected DSBs may persist in the *rad51* Δ mutant, the cells are eventually able to either repair those DSBs by SSA or adapt and extinguish the checkpoint.

Text added in lines 455-458: The surviving *rad51* Δ mutant cells can extinguish the DNA damage checkpoint by 24 hours post-DSB induction (Supplementary Fig. 6i), though this is delayed compared to the 12 hours observed in wildtype cells (Supplementary Fig. 1d).

On this turn, how is the double mutant *rad9 rad51* in term of vitality and repeat contraction/expansion?

We have performed this experiment and added it to Fig. 6a,b. We found that we lose the *rad9* Δ -dependent rescue in the absence of Rad51 suggesting that Rad51 functions upstream of Rad9 in resection/gap-filling: in the absence of Rad51 to protect the resected DNA and/or facilitate BIR-repair, faster resection in the *rad9* Δ mutant cannot rescue inviability due to breaks.

Text added in lines 405-412: Deletion of *RAD51* in a *rad9* Δ mutant resulted in an even lower level of viability in both the scrm(CTG)₇₀ and (CTG)₇₀ template strains compared to the *rad51* Δ single mutant, eliminating the rescue observed in the *rad9* Δ mutant (Fig. 6a). Similarly, CTG contractions were still high in the *rad9* Δ *rad51* Δ double mutant with no rescue (Fig. 6b). Therefore, Rad51 functions upstream of Rad9 in gap repair. We conclude that in the absence of Rad51 to protect the resected DNA and/or facilitate

BIR-repair, faster resection in the *rad9Δ* mutant cannot rescue repeat contractions due to template hairpin formation or inviability due to breaks.

Is it possible to show the terminal phenotype of *rad51* cells, following HO break? Do they remain blocked in G2/M (similar to adaptation-defective cells) or start dividing forming microcolonies of few cells?

In addition to measuring checkpoint activation (discussed above), we also performed a microcolony experiment with the (CTG)₇₀ template wildtype and *rad51Δ* mutant cells (Supplementary Fig. 6h). We found that wildtype cells are able to undergo several divisions after 24 hours on galactose. However, for the *rad51Δ* mutant 60% of cells were able to undergo cell division but the remaining 40% of cells were still arrested at the G2/M phase. Presumably those cells that have undergone cell division in the *rad51Δ* mutant were able to adapt (as also seen by Western blot) or repair by some Rad51-independent mechanism (such as SSA) as observed by appearance of the U2 repair product by Southern blot.

Text added in lines 450-455: Given the decreased viability and reduced healing in the *rad51Δ* mutant, we asked whether it has a defect in resumption of cell division post-DSB induction in the (CTG)₇₀ template strain. Following *rad51Δ* mutant cells over 24 hours showed that only 14% of cells were able to complete more than 5 divisions, suggesting some *rad51Δ* mutant cells can eventually either heal or adapt, but many are permanently arrested (Supplementary Fig. 6h).

Reviewer #3 (Remarks to the Author):

The manuscript provides new mechanistic insights into triplet repeat instability during repair of large DNA gaps in yeast. Some of the findings, such as contraction bias of CTG template and expansion bias of CAG templates, had been inferred in earlier studies, but the manuscript does break new ground by testing specific models of instability and looking at DNA intermediates during processing. For example, the data in Fig3b are especially interesting because the expansion bias seen under DSB conditions mimics to some extent what is seen in long triplet repeat tracts in humans and mice. The second cut site in single stranded CTG tracts is also novel and interesting. The interpretation of these data provides some insights into mechanistic behavior of the DNA strands and their interaction with repair proteins that might apply, in general, to mammals as well as yeast.

The one major issue with the manuscript is the control experiment looking for hyperactivation of Rad53. Lines 123-125 claim there was no persistent hyperactivation of Rad53, but Fig S1c is hard to interpret due to frowning on the gel.

We have repeated all Westerns in Supplementary Fig.1c and added a positive control. Our new results are of higher quality with no frowning and allow a clearer comparison to the positive control. We still see no differences in Rad53 hyper-phosphorylation with or

without a CTG repeat tract. Our conclusion remains that there is no difference in checkpoint activation or extinguishment in strains that have a repeat tract compared to strains that do not.

Also, the figure does not provide a positive control for phosphoRad53. Better data is required to confirm this key control.

We have added a control lane to all Rad53 phosphorylation westerns where the parental strain was treated with 0.035% MMS for 90 minutes and then protein was harvested.

Minor points

1. Lines 133-134 claim that resection, annealing and fill-in synthesis are likely occurring “concurrently”. This is an overstatement since, from the data presented, as much as two hours may be elapsing between resection and fill-in.

Although there is overlap in the two events, we agree that concurrently was the wrong term to use. This sentence was removed.

2. The Rad51 enrichment data in SFig2b are a bit confusing in light of the earlier claims of the paper that the assay is SSA (abstract) but that SSA is Rad51-independent (lines 139-140). A bit more explanation is needed to make this point more clear at this point of the manuscript. The SSA-BIR description later is helpful, but some clarity in the text about SFig2b is needed.

We have added the following clarifying remarks:

Ln 149-154: In addition, Rad51 is recruited to ssDNA to initiate the formation of the nucleoprotein filament necessary for the homology search¹¹. Though the resected break in this assay system predominantly repairs via SSA it can also repair via BIR which is Rad51-dependent¹¹. With the expectation that RPA and Rad51 enrichment would increase during resection and then decrease during gap filling, we monitored enrichment of RPA and Rad51 in both the (CTG)₇₀ and scrm(CTG)₇₀ strains.

3. Line 253 mentions dark green boxes in Fig 4a, but I think this should read purple boxes.

This notation has been fixed.

4. Lines 431-4 describe experiments from HD mice (ref 6). A similar idea was proposed for myotonic dystrophy based on mathematical modelling (10.1093/hmg/dds059). It would be useful to consider citing the DM paper in addition.

Thank you for this excellent reference. We have added it to our discussion lines 505-511, which reads:

Modeling of the expansion bias observed in humans using a large sample of blood DNA samples collected from Myotonic Dystrophy type 1 (DM1) patients predicted that hundreds of small expansion and contraction events accumulate over the lifespan of a hematopoietic stem cell, and the rate was consistent with occurrence due to DNA damage and repair rather than replication⁷. Thus, our data provide a mechanism for small TNR expansion events that can result in worsening of disease phenotypes in somatic cells.

References (response only)

- 1 Zhu, Z., Chung, W. H., Shim, E. Y., Lee, S. E. & Ira, G. Sgs1 helicase and two nucleases Dna2 and Exo1 resect DNA double-strand break ends. *Cell* **134**, 981-994, doi:10.1016/j.cell.2008.08.037 (2008).
- 2 Mimitou, E. P. & Symington, L. S. Sae2, Exo1 and Sgs1 collaborate in DNA double-strand break processing. *Nature* **455**, 770-774, doi:10.1038/nature07312 (2008).
- 3 Gravel, S., Chapman, J. R., Magill, C. & Jackson, S. P. DNA helicases Sgs1 and BLM promote DNA double-strand break resection. *Genes Dev* **22**, 2767-2772, doi:10.1101/gad.503108 (2008).
- 4 Ferrari, M. *et al.* Functional interplay between the 53BP1-ortholog Rad9 and the Mre11 complex regulates resection, end-tethering and repair of a double-strand break. *PLoS Genet* **11**, e1004928, doi:10.1371/journal.pgen.1004928 (2015).
- 5 Erwin, G. S. *et al.* Recurrent repeat expansions in human cancer genomes. *Nature* **613**, 96-102, doi:10.1038/s41586-022-05515-1 (2023).
- 6 Polyzos, A. A. & McMurray, C. T. Close encounters: Moving along bumps, breaks, and bubbles on expanded trinucleotide tracts. *DNA Repair (Amst)* **56**, 144-155, doi:10.1016/j.dnarep.2017.06.017 (2017).
- 7 Higham, C. F., Morales, F., Cobbold, C. A., Haydon, D. T. & Monckton, D. G. High levels of somatic DNA diversity at the myotonic dystrophy type 1 locus are driven by ultra-frequent expansion and contraction mutations. *Hum Mol Genet* **21**, 2450-2463, doi:10.1093/hmg/dds059 (2012).
- 8 Mosbach, V. *et al.* Resection and repair of a Cas9 double-strand break at CTG trinucleotide repeats induces local and extensive chromosomal deletions. *PLoS Genet* **16**, e1008924, doi:10.1371/journal.pgen.1008924 (2020).

- 9 Freudenreich, C. H., Kantrow, S. M. & Zakian, V. A. Expansion and length-dependent fragility of CTG repeats in yeast. *Science* **279**, 853-856 (1998).
- 10 Saini, N. *et al.* Migrating bubble during break-induced replication drives conservative DNA synthesis. *Nature* **502**, 389-392, doi:10.1038/nature12584 (2013).
- 11 Jain, S. *et al.* A recombination execution checkpoint regulates the choice of homologous recombination pathway during DNA double-strand break repair. *Genes Dev* **23**, 291-303, doi:10.1101/gad.1751209 (2009).
- 12 Lazzaro, F. *et al.* Histone methyltransferase Dot1 and Rad9 inhibit single-stranded DNA accumulation at DSBs and uncapped telomeres. *EMBO J* **27**, 1502-1512, doi:10.1038/emboj.2008.81 (2008).
- 13 Chen, H., Lisby, M. & Symington, L. S. RPA coordinates DNA end resection and prevents formation of DNA hairpins. *Mol Cell* **50**, 589-600, doi:10.1016/j.molcel.2013.04.032 (2013).
- 14 Giaccherini, C. & Gaillard, P. H. Control of structure-specific endonucleases during homologous recombination in eukaryotes. *Curr Opin Genet Dev* **71**, 195-205, doi:10.1016/j.gde.2021.09.005 (2021).
- 15 Young, S. J. & West, S. C. Coordinated roles of SLX4 and MutSbeta in DNA repair and the maintenance of genome stability. *Crit Rev Biochem Mol Biol* **56**, 157-177, doi:10.1080/10409238.2021.1881433 (2021).
- 16 Ait Saada, A. *et al.* Structural parameters of palindromic repeats determine the specificity of nuclease attack of secondary structures. *Nucleic Acids Res* **49**, 3932-3947, doi:10.1093/nar/gkab168 (2021).
- 17 Clerici, M., Mantiero, D., Lucchini, G. & Longhese, M. P. The *Saccharomyces cerevisiae* Sae2 protein promotes resection and bridging of double strand break ends. *J Biol Chem* **280**, 38631-38638, doi:10.1074/jbc.M508339200 (2005).
- 18 Lobachev, K. S., Gordenin, D. A. & Resnick, M. A. The Mre11 complex is required for repair of hairpin-capped double-strand breaks and prevention of chromosome rearrangements. *Cell* **108**, 183-193, doi:10.1016/s0092-8674(02)00614-1 (2002).
- 19 Chen, X. *et al.* The Fun30 nucleosome remodeller promotes resection of DNA double-strand break ends. *Nature* **489**, 576-580, doi:10.1038/nature11355 (2012).
- 20 Ferrari, M., Rawal, C. C., Lodovichi, S., Vietri, M. Y. & Pelliccioli, A. Rad9/53BP1 promotes DNA repair via crossover recombination by limiting the Sgs1 and Mph1 helicases. *Nat Commun* **11**, 3181, doi:10.1038/s41467-020-16997-w (2020).

- 21 Khristich, A. N. & Mirkin, S. M. On the wrong DNA track: Molecular mechanisms of repeat-mediated genome instability. *J Biol Chem* **295**, 4134-4170, doi:10.1074/jbc.REV119.007678 (2020).

REVIEWERS' COMMENTS

Reviewer #1 (Remarks to the Author):

In this revised manuscript, the authors addressed most of my concerns. However, there are a few places that confused me.

1. Page 7, lines 154-156, the sentence "Indeed, in both strains, maximum enrichment of RPA and Rad51 60 bp and 600 bp after the repeat occurs 6 hours after DSB induction when ssDNA is maximal." is confusing. Please clarify.
2. Page 8, line 163, "...which was significant 8 hours post-DSB induction". What does the significant 8 hours mean? Please clarify.
3. Line 256, 369 should "gap fill-in" be "gap filling"? Please double check it to make it consistent.
4. Line 259, " polymerase fill-in" should be "polymerase gap filling".

Reviewer #2 (Remarks to the Author):

The authors have done excellent work to address my concerns. I understand their arguments regarding the connection with the DNA resection metabolism, and I agree that this part can be better investigated by future studies without compromising present findings. Overall, the revised manuscript improved both in text and experiments.

Reviewer #3 (Remarks to the Author):

The manuscript provides new mechanistic insights into triplet repeat instability during repair of large DNA gaps in yeast. Some of the findings, such as contraction bias of CTG template and expansion bias of CAG templates, had been inferred in earlier studies, but the manuscript does break new ground by testing specific models of instability and looking at DNA intermediates during processing. For example, the data in Fig3b are especially interesting because the expansion bias seen under DSB conditions mimics to some extent what is seen in long triplet repeat tracts in humans and mice. The second cut site in single stranded CTG tracts is also novel and interesting. The interpretation of these data provides some insights into mechanistic behavior of the DNA strands and their interaction with repair proteins that might apply, in general, to mammals as well as yeast.

The revised manuscript satisfies my concerns.

Second Resubmission

REVIEWERS' COMMENTS

Manuscript # NCOMMS-22-11274A

Reviewer #1 (Remarks to the Author):

In this revised manuscript, the authors addressed most of my concerns. However, there are a few places that confused me.

1. Page 7, lines 154-156, the sentence “Indeed, in both strains, maximum enrichment of RPA and Rad51 60 bp and 600 bp after the repeat occurs 6 hours after DSB induction when ssDNA is maximal.” is confusing. Please clarify.

Ln 150-155: Sentences have been edited to: With the expectation that RPA and Rad51 enrichment would increase during resection and then decrease during gap filling, we monitored enrichment of RPA and Rad51 in both the (CTG)₇₀ and scrm(CTG)₇₀ strains. Monitoring enrichment of RPA and Rad51 either 60 bp or 600 bp after the repeat tract showed increasing enrichment of both proteins up until 6 hours post DSB induction and then a subsequent decrease in enrichment, mirroring the timing profile of resection and fill-in.

2. Page 8, line 161, “...which was significant 8 hours post-DSB induction”. What does the significant 8 hours mean? Please clarify.

Sentence has been clarified to (Ln 161-164): Even though there is no repair defect in the (CAG)₇₀ template strain (Fig. 2a, b), a decrease in ssDNA accumulation post-DSB induction was observed in this strain compared to the scrambled control, indicating that less ssDNA accumulates beyond the repeat tract when (CTG)₇₀ is on the resected strand (Fig. 2c).

3. Line 257, 370 should “gap fill-in” be “gap filling”? Please double check it to make it consistent.

Ln 257: Line should read gap fill-in-dependent expansions as it modifies the word expansions (extra hyphen added).

Ln 370: word usage has been fixed, sentence reads: Consistent with faster resection and gap filling, there was a significantly earlier appearance of the U2 repair product in the *rad9Δ* mutant compared to wildtype (Fig. 5d, Supplementary Fig. 5b).

4. Line 259, " polymerase fill-in" should be "polymerase gap filling".

Word usage has been fixed.

Reviewer #2 (Remarks to the Author):

The authors have done excellent work to address my concerns. I understand their arguments regarding the connection with the DNA resection metabolism, and I agree that this part can be better investigated by future studies without compromising present findings. Overall, the revised manuscript improved both in text and experiments.

Reviewer #3 (Remarks to the Author):

The manuscript provides new mechanistic insights into triplet repeat instability during repair of large DNA gaps in yeast. Some of the findings, such as contraction bias of CTG template and expansion bias of CAG templates, had been inferred in earlier studies, but the manuscript does break new ground by testing specific models of instability and looking at DNA intermediates during processing . For example, the data in Fig3b are especially interesting because the expansion bias seen under DSB conditions mimics to some extent what is seen in long triplet repeat tracts in humans and mice. The second cut site in single stranded CTG tracts is also novel and interesting. The interpretation of these data provides some insights into mechanistic behavior of the DNA strands and their interaction with repair proteins that might apply, in general, to mammals as well as yeast.

The revised manuscript satisfies my concerns.